

# Boundary operator expansion and extraordinary phase transition in the tricritical O(N) model

**Xinyu Sun[1] and Shao-Kai Jian[2]⋆**

**1** Institute for Advanced Study, Tsinghua University, Beijing 100084, China
**2** Department of Physics and Engineering Physics, Tulane University,
New Orleans, Louisiana, 70118, USA

⋆ sjian@tulane.edu

## Abstract

We study the boundary extraordinary transition of a three-dimensional (3D) tricritical $O(N)$ model. We first compute the mean-field Green's function with a general coupling of $|\vec{\phi}|^{2n}$ (with $n = 3$ corresponding to the tricritical model) at the extraordinary phase transition. Then, using layer susceptibility, we obtain the boundary operator expansion for the transverse and longitudinal modes within the $\epsilon = 3 - d$ expansion. Based on these results, we demonstrate that the tricritical point exhibits an extraordinary transition characterized by an ordered boundary for any $N$. This provides the first nontrivial example of continuous symmetry breaking in 2D in the context of boundary criticality.



# 1 Introduction

Boundary conformal field theory (BCFT) is a vibrant area of research with diverse applications across multiple disciplines [1]. The presence of a boundary enriches the bulk CFT by introducing additional conformal data [2–6]. For example, in BCFTs, bulk primary operators have a bulk-to-boundary operator product expansion (OPE) in addition to the standard bulk-to-bulk OPE. As a result, the two-point function in BCFT plays an important role, similar to the four-point function in CFT, as it admits different OPE channels, via a bulk OPE or via a boundary operator expansion (BOE) [4, 7]. Note that this crossing symmetry is also the building block for conformal bootstrap in BCFTs [6].

A classic example of BCFT arises in the critical $O(N)$ model, where the introduction of a boundary gives rise to the ordinary, special, and extraordinary transitions [3, 7–10]. Recently, the renewed interest in the extraordinary transition in three-dimensional (3D) $O(N)$ BCFT has been driven by the discovery of the extraordinary-log transition [11]. The BOE predicted the existence of a critical number of components, $N_c$, such that a logarithmic correlation function emerges at the boundary during the extraordinary transition when $2 \leq N < N_c$. It has been later examined via Monte Carlo simulations [12–14] and the conformal bootstrap [15].

However, beyond the critical $O(N)$ model, much remains unexplored. In this paper, we aim to extend the understanding of extraordinary transitions in 3D BCFTs and investigate, in particular, continuous symmetry breaking in two dimensions in the context of boundary criticality. To achieve this, we compute the mean-field order parameter profile and propagator in the $O(N)$ model with a general $|\vec{\phi}|^{2n}$ coupling, providing a foundation for calculating the BOE in more general models. In the Appendix D, we rederive the BOE at the extraordinary transition in the critical $O(N)$ model, demonstrating the validity and versatility of our approach. To go beyond the critical $O(N)$ model, we focus on the extraordinary transition for $n = 3$, corresponding to the tricritical $O(N)$ model, the second most prevalent universality class in the $O(N)$ model [16–24]. A significant difference between the tricritical $O(N)$ model and the

critical $O(N)$ model is that the $|\vec{\phi}|^6$ interaction at $d = 3$ is marginal for the tricritical $O(N)$ model, which substantially affects the renormalization group (RG) flow and boundary behavior. Specifically, we derive the BOE at the one-loop order by calculating the layer susceptibility, revealing the boundary operator spectrum at the extraordinary transition in the tricritical $O(N)$ model.

At the extraordinary-log transition in the critical $O(N)$ model, although the logarithmic correlation decays more slowly than any power law, the surface order parameter ultimately vanishes at the infrared (IR) fixed point. While Mermin-Wagner-Coleman (M-W-C) theorem forbids spontaneous breaking of continuous symmetries in strict 2D systems, it remains a question in the context of BCFT, where the 2D boundary couples to a 3D bulk theory. Remarkably, Cuomo and Zhang recently proved a no-go theorem that forbids continuous symmetry breaking in BCFTs [25]. By analyzing the Ward identity associated with a continuous internal symmetry, they decomposed it into bulk and boundary contributions. Crucially, under the assumption that the RG flow reaches a conformal fixed point, they showed that spontaneous symmetry breaking would necessitate a pole in the boundary spectral function, corresponding to a decoupled sector of gapless Goldstone modes, which is ruled out by the M-W-C theorem. Therefore, continuous symmetry breaking is forbidden in BCFTs under this assumption.

In contrast, our analysis demonstrates that the boundary and bulk RG flows must be treated on equal footing in the tricritical $O(N)$ model, rather than assuming the bulk theory sits at an IR conformal fixed point. In three dimensions, the marginal logarithmic flow of the bulk coupling constant intrinsically influences the RG behavior of the boundary theory. As a result, we find that the boundary order parameter remains finite at the extraordinary transition of the tricritical $O(N)$ model for any $N$ in $d = 3$, in stark contrast to the vanishing surface order at the extraordinary-log transition in the critical $O(N)$ model.

On a technical note, our calculation of the BOE utilizes the one-to-one correspondence between the layer susceptibility and the Green's function [4, 26, 27]. More specifically, in the BOE channel, evaluating the two-point function can be simplified by transforming it into the layer susceptibility [28–30]. The layer susceptibility is more straightforward to compute perturbatively using Feynman diagrams, as it involves fewer intricate integrals. This approach is particularly advantageous at the extraordinary transition [31–34], where even the mean-field propagator becomes challenging to handle.

The paper is organized as follows. In the remainder of the introduction, we present a summary of our results, including the mean-field correlation function for general models and the BOE at the one-loop level for the tricritical $O(N)$ model. Section 2 provides the details of the evaluation of the mean-field correlation functions in the $O(N)$ field theory with a general $|\vec{\phi}|^{2n}$ coupling. In Section 3, we present a detailed evaluation of layer susceptibility, including the transverse and longitudinal parts, at the one-loop level via the $d = 3 - \epsilon$ calculation. With the layer susceptibility, the BOE is obtained by series expansion of the hypergeometric function in Section 4, where the boundary operator content is discussed. Section 5 gives a brief review of the extraordinary-log transition. Importantly, we show that at strict three dimensions, because of the flow of coupling $s$, the boundary remains ordered at the extraordinary transition for all $N$ in the tricritical $O(N)$ model. We conclude in Section 6 by discussing possible applications for our theory. Several appendices provide more details in the technical calculations. We also apply our general method to the critical $O(N)$ model, demonstrating its validity by rederiving the BOE.

## 1.1 Summary of the results

We consider the $O(N)$ model with a general coupling defined by the action,

$$S = \int_{z \geq 0} \mathrm{d}^{d-1} r \, \mathrm{d}z \left[ \frac{1}{2} |\nabla \vec{\phi}|^2 + \frac{u_0}{(2n)!} |\vec{\phi}|^{2n} + \frac{c_0}{2} |\vec{\phi}|^2 \delta(z) \right], \tag{1}$$

where $z$ denotes the coordinate perpendicular to the boundary at $z = 0$, while $r$ denotes the coordinate parallel to the boundary. $u_0$ denotes the strength of the self-coupling and $c_0$ denotes the mass on the boundary. $\vec{\phi} = (\phi_1, \ldots, \phi_N)$ is an $O(N)$ field and $|\vec{\phi}|^{2n} = \left( \sum_i \phi_i^2 \right)^n$. As we are interested in the extraordinary transition, $c_0 \to -\infty$, and the order parameter does not vanish at the mean-field level, we assume the ordered component to be $\phi_1$ without loss of generality. We consider the connected two-point functions of the order parameter field, which are separated into the transverse part, $\langle \phi_i(r,z) \phi_j(r',z') \rangle_{\mathrm{con}} = \delta_{ij} G^T(r - r', z, z')$ with no summation over $i > 1$, and the longitudinal part, $\langle \phi_1(r,z) \phi_1(r',z') \rangle_{\mathrm{con}} = G^L(r - r', z, z')$. Our contributions include the derivation of the mean-field correlation function for this general model, the layer susceptibility, and the BOE at the one-loop level for $n = 3$, corresponding to the tricritical $O(N)$ model. Additionally, we establish the presence of the extraordinary phase transition for arbitrary $N$ in the tricritical $O(N)$ model characterized by an ordered boundary in $d = 3$, in stark contrast to the critical $O(N)$ theory. Below, we summarize our results.

### 1.1.1 Mean-field correlation function for the $O(N)$ model with $|\vec{\phi}|^{2n}$ coupling

The mean-field order parameter profile, $m_0(z) \equiv \langle \phi_1(r,z) \rangle_0$ is

$$m_0(z) = \alpha (z + \beta)^{-1/(n-1)}, \tag{2}$$

with

$$\alpha = \left( \frac{(2n-1)! \cdot n}{(n-1)^2 u_0} \right)^{\frac{1}{2n-2}}, \qquad \beta = -\frac{1}{(n-1)c_0}. \tag{3}$$

Because $c_0 < 0$ at the extraordinary transition, $\beta > 0$ and the order parameter profile is well-defined for $z \geq 0$. Moreover, $m_0(z \to \infty) = 0$ as expected for a vanishing order deep in the bulk. At the extraordinary fixed point, one can set $c_0 \to -\infty$, and use $\beta = 0$ to simplify the result.

The mean-field two-point function in the mixed representation can be expressed by the modified Bessel function $K_\nu(x)$ and $I_\nu(x)$:

$$G_0^{L,T}(p,z,z') = \begin{cases} \sqrt{zz'} I_{\alpha_{L,T}}(pz') K_{\alpha_{L,T}}(pz), & z > z', \\ \sqrt{zz'} K_{\alpha_{L,T}}(pz') I_{\alpha_{L,T}}(pz), & z < z', \end{cases} \tag{4}$$

where $p = |\vec{p}|$ is the momentum parallel to the boundary surface, and $\alpha_L = \alpha_T + 1 = \frac{3}{2} + \frac{1}{n-1}$. From the two-point function, the mean-field layer susceptibility can be obtained via Eq. (26) as

$$\chi_0^{L,T}(z,z') = \sqrt{zz'} \frac{\zeta^{\alpha_{L,T}}}{2\alpha_{L,T}}, \qquad \zeta = \frac{\min(z,z')}{\max(z,z')}. \tag{5}$$

More generally, the layer susceptibility in BCFT admits the following expansion,

$$\chi(z,z') = (4zz')^{\frac{d-1}{2} - \Delta_\phi} \sum_{\Delta > 0} c_\Delta \zeta^{\Delta - \frac{d-1}{2}}, \tag{6}$$

where $\Delta_\phi$ denotes the scaling dimension of the bulk primary field while $\Delta$ refers to the scaling dimension of the boundary primary fields. The expansion coefficient, $c_\Delta$, and the operator spectrum, $\Delta$, are universal BCFT data. In the mean-field theory, the scaling dimension of the $O(N)$ field is $\Delta_\phi = \frac{d-2}{2}$. From the mean-field layer susceptibility, one can see that the scaling dimensions of the boundary longitudinal and transverse boundary modes are $\Delta_{L,T} = \frac{d-1}{2} + \alpha_{L,T}$, and the corresponding expansion coefficients, $c_\Delta = \frac{1}{4\alpha_{L,T}}$, respectively. In the following, we focus on the tricritical $O(N)$ model and present results at the one-loop level.

### 1.1.2 Boundary operator expansion in the tricritical $O(N)$ model

We focus on the tricritical $O(N)$ model with $n = 3$. The order parameter profile at the one-loop order is

$$m(z) = m_0(z) + m_1(z) = \left(\frac{90}{u_0}\right)^{\frac{1}{4}} z^{-\frac{1}{2}} \left[ 1 - \sqrt{\frac{5u_0}{8}} \frac{\frac{1+d}{5-d} + \frac{N-1}{5}}{(4-d)d} \gamma_T z^{3-d} \right], \tag{7}$$

with $\gamma_T \equiv \frac{\Gamma(\frac{d+1}{2})\Gamma(\frac{2-d}{2})}{2^d \pi^{\frac{d}{2}} \Gamma(\frac{5-d}{2})}$. Notice that $\gamma_T < 0$ for $2 < d < 4$. The normalization of the one-point function can be obtained at the one-loop level,

$$m(z) = \frac{a_\sigma}{(2z)^{1/2}}, \qquad a_\sigma \approx \left(\frac{3(3N+22)}{8\pi^2}\right)^{\frac{1}{4}} \epsilon^{-\frac{1}{4}} + \frac{9+N}{2} \left(\frac{1}{6(3N+22)\pi^2}\right)^{\frac{1}{4}} \epsilon^{\frac{1}{4}}, \tag{8}$$

where $\epsilon = 3-d$ and the fixed point of the coupling in Eq. (54) has been used. At strictly $d = 3$, if we naively take $\epsilon = 0$ in $a_\sigma$, we would get an unphysical divergence for the order parameter. Instead, we should consider the logarithmic flow of the coupling in Eq. (60), which in turn leads to

$$m(z) \sim \frac{(\log \mu_0 z)^{1/4}}{z^{1/2}}, \tag{9}$$

as expected at the upper critical dimension: the order parameter vanishes in the deep bulk limit, $z \to \infty$, with a logarithmic correction. Here, $\mu_0$ is an arbitrary energy scale.

Now, we present the BOE of the layer susceptibility at the one-loop level. The transverse layer susceptibility reads,

$$\chi^T(z_1, z_2) = \sqrt{4z_1 z_2} \zeta^{-\frac{d-1}{2}} \sum_{\Delta = d-1, k} c_\Delta^T \zeta^\Delta, \tag{10}$$

where the expansion coefficients are given by

$$c_{d-1}^T = \frac{1}{4} - \frac{2 + 3c_{1,1} - 3c_{2,1}}{128\pi} \sqrt{\frac{u_0}{10}} = \frac{1}{4} - \frac{2 + 3c_{1,1} - 3c_{2,1}}{16} \sqrt{\frac{3}{2(3N+22)}} \epsilon^{\frac{1}{2}}, \tag{11}$$

$$c_k^T = \frac{3}{8\pi} \frac{16}{k^2(k-1)(k-2)^2} \sqrt{\frac{u_0}{10}} = \frac{3}{2} \sqrt{\frac{6}{3N+22}} \frac{1}{k^2(k-1)(k-2)^2} \epsilon^{\frac{1}{2}}, \quad k = 5, 7, 9, \dots, \tag{12}$$

where $c_{1,1}, c_{2,1}$ are given by complicated series, but their numerical values can be obtained as $c_{1,1} \approx -1.39608$ and $c_{2,1} \approx -1.48791$.

The primary boundary operator with scaling dimension of $d - 1$ at both the leading and subleading order in the first line is related to the $O(N)$ conserved current that is broken at the boundary, i.e., the tilt field. It is also worth noting that in contrast to the critical $\phi^4$ theory, the leading term in the $\epsilon$ expansion is $\mathcal{O}(\epsilon^{1/2})$. This is not a coincidence; the behavior of $\mathcal{O}(\epsilon^{1/2})$ also appears at the special transition in the tricritical model [22].

The layer susceptibility for the longitudinal component is

$$\chi^L(z_1, z_2) = \sqrt{4z_1 z_2}\,\zeta^{-\frac{d-1}{2}} \sum_{\Delta = d,k} c_\Delta^L \zeta^\Delta. \tag{13}$$

with the following expansion coefficient,

$$
\begin{aligned}
c_d^L &= \frac{1}{8} + \frac{16 + 405(c'_{1,1} - c'_{2,1}) + 15(N-1)(c''_{1,1} - c''_{2,1})}{640} \frac{1}{8\pi}\sqrt{\frac{u_0}{10}}, \\
&= \frac{1}{8} + \frac{16 + 405(c'_{1,1} - c'_{2,1}) + 15(N-1)(c''_{1,1} - c''_{2,1})}{640}\sqrt{\frac{3}{2(3N+22)}}\epsilon^{\frac{1}{2}} + \mathcal{O}(\epsilon),
\end{aligned}
\tag{14}
$$

where the numerical values are $c'_{1,1} \approx -1.45699$, $c'_{2,1} \approx -1.35775$ and $c''_{1,1} \approx -1.64248$, $c''_{2,1} \approx -0.263237$. The operator with scaling dimension $d$ corresponds to the displacement operator. The presence of a boundary breaks the translation symmetry. As a result, the conservation of stress-energy tensor is violated by a boundary term, i.e, the displacement operator, whose scaling dimension is protected to be $d$. Besides the displacement field, the expansion coefficients for other boundary operators appeared in the BOE are given by

$$
\begin{aligned}
c_4^L &= \frac{N-1}{400\pi}\sqrt{\frac{u_0}{10}} = \sqrt{\frac{3}{2(3N+22)}} \cdot \frac{N-1}{50}\epsilon^{\frac{1}{2}} + \mathcal{O}(\epsilon), \\
c_k^L &= \frac{3}{16\pi} \frac{N+26}{(k-3)^2(k-1)(k+1)^2}\sqrt{\frac{u_0}{10}} \\
&= \frac{3}{2}\sqrt{\frac{3}{2(3N+22)}} \cdot \frac{N+26}{(k-3)^2(k-1)(k+1)^2}\epsilon^{\frac{1}{2}} + \mathcal{O}(\epsilon), \quad k = 6,8,10,\dots
\end{aligned}
\tag{15}
$$

Again, we see that the leading term in the $\epsilon$ expansion is $\mathcal{O}(\epsilon^{1/2})$. Also, note that the BOE coefficient $c_4^L$ vanishes when $N = 1$. The prefactor originates from a loop diagram formed by propagators of the transverse modes, yielding the $N-1$ factor. Hence, for $N = 1$, this contribution is absent because there are no transverse modes.

Naively, setting $\epsilon = 0$ in the final expression for the expansion coefficients at $d = 3$ appears to suggest that BOE vanishes. However, in strictly $d = 3$, $u_0$ exhibits a logarithmic flow, given by $u_0 \sim \frac{1}{\log \mu_0 z}$. Substituting this logarithmic behavior into the expression of BOE with explicit $u_0$ dependence in the first equality reveals that BOE acquires a logarithmic dependence instead of being completely trivial.

### 1.1.3 Extraordinary phase transition in the tricritical $O(N)$ model

Lastly, we discuss the extraordinary transition. To investigate the boundary order, we consider a coupled theory involving the normal fixed point action and the nonlinear sigma model (NLSM):

$$S_{\mathrm{IR}} = S_{\mathrm{normal}} + S_n - s \int \mathrm{d}^{d-1}r \sum_{i>1} \pi_i(r)\hat{t}_i(r), \tag{16}$$

$$S_n = \int \mathrm{d}^{d-1}r \frac{1}{2g}(\partial_\mu \vec{n})^2, \tag{17}$$

where $\vec{n} = (\sqrt{1 - \vec{\pi}^2}, \vec{\pi})$ and $\hat{t}_i$ is the tilt field. $s$ denotes the bulk-boundary coupling and $g$ denotes the coupling in the NLSM. The corresponding coupled RG equations are

$$\frac{\mathrm{d}s}{\mathrm{d}\log\mu} = -\frac{3(3N+22)}{128\pi^4}s^{-3}, \tag{18}$$

$$\frac{\mathrm{d}g}{\mathrm{d}\log\mu} = -\frac{N-2}{2\pi}g^2 + \frac{\pi s^2}{2}g^2, \tag{19}$$



with the solution in low energies being,

$$s \approx \left[ \frac{3(3N+22)}{2(2\pi)^4} \log \mu_0 l \right]^{1/4}, \quad g \approx \frac{48(N-2)\pi}{3N+22} (\log \mu_0 l)^{-2} + \sqrt{\frac{96\pi^2}{3N+22}} (\log \mu_0 l)^{-3/2}, \quad (20)$$

where $\mu_0$ is an arbitrary energy scale, and $l$ denotes the length scale of the system. Notice that the logarithmic flow of coupling $g$ differs from that in the critical $O(N)$ model: it decays more rapidly than $(\log \mu_0 l)^{-1}$ at the long-wavelength limit. Consequently, we have the expectation value of the boundary order parameter at low energies,

$$\langle n \rangle_r \sim \exp \left[ (N-1) \sqrt{\frac{24}{3N+22}} \left( \log \frac{\mu_0}{\sqrt{h}} \right)^{-1/2} \right], \quad h \to 0, \quad (21)$$

where $h$ represents an external field linearly coupled to the boundary order $\vec{n}$. We can see that, as the external field $h$ decreases to zero, the expectation of the boundary order remains nonzero, with $\langle n \rangle_r \to 1$, rather than vanishing. Consequently, the tricritical point exhibits an extraordinary transition with an ordered boundary, in stark contrast to that in the critical $O(N)$ model. This can be understood intuitively, as the upper critical dimension for the tricritical point is $d = 3$, resulting in milder fluctuations compared to those at the critical $O(N)$ point. This provides a nontrivial example of continuous symmetry breaking in two dimensions within the context of the boundary criticality.

## 2 Mean-field correlation functions at extraordinary transition

We consider the $O(N)$ model in a $d$-dimensional semi-infinite space, where the bulk corresponds to $z > 0$ and the boundary lies at $z = 0$. In this section, we present the mean-field Green's functions for the $O(N)$ model with general interactions, defined by the following action

$$S = \int_{z \geq 0} d^{d-1} r \, dz \left( \frac{1}{2} |\nabla \vec{\phi}|^2 + \frac{u_0}{(2n)!} |\vec{\phi}|^{2n} + \frac{c_0}{2} |\vec{\phi}|^2 \delta(z) \right). \quad (22)$$

Here, $z$ denotes the coordinate perpendicular to the boundary surface at $z = 0$, while $r$ denotes the coordinate parallel to the boundary. $\vec{\phi} = (\phi_1, \ldots, \phi_N)$ is an $O(N)$ vector field and $u_0$ denotes the self-coupling strength. Note that we consider a general coupling of the order $2n$. $c_0$ is the boundary mass term. In general, boundary self-couplings should be included. For instance, a boundary fourth order self-coupling should be included at the special and ordinary transitions in the tricritical $O(N)$ model. However, we omit the boundary self-coupling in this section for simplicity. More importantly, we can show that all boundary self-couplings are irrelevant at the extraordinary transition of the tricritical $O(N)$ model.

Before proceeding, we briefly summarize the properties of the BOE in CFTs. Conformal symmetry implies that the BOE of two-point functions can be decomposed in terms of boundary conformal blocks. Consider the two-point function of a primary field. It is well known that it can be expressed as follows [4, 6, 27],

$$G(r - r', z, z') = (4zz')^{-\Delta_\phi} \sum_{\Delta > 0} c_\Delta \sigma_\Delta \mathcal{G}_{\text{boe}}(\Delta, \xi), \quad (23)$$

$$\sigma_\Delta = 4^{-\Delta + \frac{d-1}{2}} \pi^{-\frac{d-1}{2}} \Gamma(\Delta) / \Gamma \left( \Delta - \frac{d-1}{2} \right), \quad \xi = \frac{(r-r')^2 + (z-z')^2}{4zz'}, \quad (24)$$

where $r, r'$ are the coordinates in the subspace parallel to the boundary, and $z, z'$ are the coordinates perpendicular to the boundary. The BOE coefficient is written as a product of $c_\Delta$ and

$\sigma_\Delta$, and $c_\Delta$ is related to the BOE in layer susceptibility, which will be introduced later. $\Delta_\phi$ denotes the scaling dimension of the primary field $\phi$, while $\Delta$ is the scaling dimension of the boundary primary operators appeared in the BOE. The boundary conformal block, uniquely fixed by the conformal symmetry, is

$$\mathcal{G}_{\text{boe}}(\Delta, \xi) = \xi^{-\Delta} \, {}_2F_1\left(\Delta, \Delta + 1 - \frac{d}{2}, 2\left(\Delta + 1 - \frac{d}{2}\right); -\xi^{-1}\right). \tag{25}$$

The layer susceptibility is defined using the connected two-point function as follows,

$$\chi(z, z') = \int d^{d-1}r \, G(r, z, z') = G(p = 0, z, z'), \tag{26}$$

where the coordinate variables in the subspace parallel to the boundary have been integrated out. Equivalently, this corresponds to the zero-momentum ($p = 0$) component of the connected two-point function in Fourier space, where $p$ denotes the momentum in directions parallel to the boundary. This formulation significantly simplifies the calculations, while still preserving the full information about the BOE of the two-point function. Specifically, the layer susceptibility admits the following expansion,

$$\chi(z, z') = (4zz')^{\frac{d-1}{2} - \Delta_\phi} \sum_{\Delta > 0} c_\Delta \zeta^{\Delta - \frac{d-1}{2}}, \qquad \zeta = \frac{\min(z, z')}{\max(z, z')}. \tag{27}$$

The relation between the BOE of the two-point function and the layer susceptibility is manifest in this expression as one can see that the BOE in layer susceptibility contains the same expansion coefficient $c_\Delta$ as well as the same operator spectrum $\Delta$. Hence, once the layer susceptibility is obtained, one is, in principle, able to extract the coefficient and the operator spectrum to construct the two-point function.

In the following, we derive the mean field order parameter profile and the mean field Green's function at the extraordinary transition. We note that while these results were derived for the $\phi^4$ model [26], the results for $n \geq 3$ are new.

## 2.1 Mean-field order parameter profile

The mean-field order parameter profile and Green's function serve as building blocks for our subsequent calculation for the layer susceptibility. We denote the order parameter profile as $m(z) \equiv \langle \phi_1(r, z) \rangle$, which does not depend on $r$ due to the translational symmetry in the hypersurface parallel to the boundary. Note that we have taken the ordered field to be $\phi_1$ without loss of generality. Via a variational principle, the mean-field equation for the order parameter is

$$-\frac{d^2}{dz^2} m_0(z) + \frac{u_0}{(2n-1)!} (m_0(z))^{2n-1} = 0, \tag{28}$$

with the boundary condition

$$\left(\frac{d}{dz} - c_0\right) m_0(z)|_{z=0} = 0. \tag{29}$$

The subscript is used to indicate the mean-field calculation. The solution is

$$m_0(z) = \alpha(z + \beta)^{-1/(n-1)}, \tag{30}$$

with

$$\alpha = \left(\frac{(2n-1)! \cdot n}{(n-1)^2 u_0}\right)^{\frac{1}{2n-2}}, \qquad \beta = -\frac{1}{(n-1)c_0}. \tag{31}$$

Because $c_0 < 0$ at the extraordinary transition and $\beta > 0$, the order parameter profile is well-defined for $z \geq 0$. Additionally, $m_0(z \to \infty) = 0$, indicating a vanishing order deep in the bulk as expected. For $n = 2$, we have $\alpha = \sqrt{\frac{12}{u}}$, $\beta = -\frac{1}{c_0}$, and $m_0(z) = \sqrt{\frac{12}{u}}\left(z + \frac{1}{|c_0|}\right)^{-1}$. Taking $c_0 \to -\infty$ recovers the result for the $\phi^4$ theory [26].

## 2.2 Mean-field two-point function

We derive the mean-field propagator based on the mean-field order parameter profile. We consider the connected two-point functions of the order parameter field, which can be separated into the transverse part,

$$\langle \phi_i(r,z)\phi_j(r',z')\rangle_{\text{con}} = \delta_{ij} G^T(r - r', z, z'), \qquad i, j > 1, \tag{32}$$

and the longitudinal part,

$$\langle \phi_1(r,z)\phi_1(r',z')\rangle_{\text{con}} = G^L(r - r', z, z'), \tag{33}$$

where $r, r'$ are the coordinates in the subspace parallel to the boundary, and $z, z'$ are the coordinates perpendicular to the boundary. The Green's function is first obtained in a mixed representation, namely, as a function of the momentum, $p$, conjugate to the coordinate parallel to the boundary and the position, $z$, away from the boundary. To proceed, we substitute the order parameter profile into the action, and then, via a variational principle, the Green's function satisfies the following differential equation and boundary condition:

$$\left(-\frac{\mathrm{d}^2}{\mathrm{d}z^2} + p^2 + 2\frac{u}{(2n)!}C_{L,T}\cdot(m_0(z))^{2n-2}\right)G_0^{L,T}(p,z,z') = \delta(z - z'),$$

$$\left(-\frac{\mathrm{d}}{\mathrm{d}z} + c_0\right)G_0^{L,T}(p,z,z')|_{z=0} = 0, \tag{34}$$

where $C_L = n(2n-1)$ and $C_T = n$ for longitudinal and transverse fields, respectively, and $m_0(z)$ is given in Eq. (30).[1] By taking $c_0 \to -\infty$, the differential equation simplifies to

$$\left[-\frac{\mathrm{d}^2}{\mathrm{d}z^2} + p^2 + \left(\alpha_{L,T}^2 - \frac{1}{4}\right)z^{-2}\right]G_0^{L,T}(p,z,z') = \delta(z - z'), \tag{35}$$

with $\alpha_L = \alpha_T + 1 = \frac{3n-1}{2(n-1)} = \frac{3}{2} + \frac{1}{n-1}$.

To solve for the Green's function, we consider the corresponding homogeneous equation and render it dimensionless using the variable $x = pz$,

$$\left[-\frac{\mathrm{d}^2}{\mathrm{d}x^2} + 1 + \left(\alpha_{L,T}^2 - \frac{1}{4}\right)x^{-2}\right]W(x) = 0. \tag{36}$$

Defining $W(x) = \sqrt{x}Q(x)$, we obtain

$$x^2 Q''(x) + xQ'(x) - \left(x^2 + \alpha_{L,T}^2\right)Q(x) = 0, \tag{37}$$

which is the standard modified Bessel equation, with two solutions, the modified Bessel function $I_\alpha(x)$ and $K_\alpha(x)$, with $\alpha = \alpha_{L,T}$. Because we require $Q(x \to \infty) = 0$ for a physical solution, the appropriate solution is $K_\alpha(x)$. Therefore, up to a constant prefactor, we have

---

[1]Explicitly, $C_L$ and $C_T$ come from the prefactors of $\phi_1^2$ and $\phi_i^2$, respectively, in the expansion of $\left((\phi_1 + m_0)^2 + \sum_{i>1}\phi_i^2\right)^n$.

$W(x) = \sqrt{x}K_\alpha(x)$.[2] By standard technique [36], the solution for the Green's function can be expressed as

$$G_0^{L,T}(p,z,z') = \begin{cases} W_0(z,p)U(z',p), & z > z', \\ U(z,p)W_0(z',p), & z < z'. \end{cases} \tag{39}$$

Since $z,z'$ are symmetric, we consider $z < z'$ without loss of generality. Thus, we have

$$U(z,p) = C_1 W_1(z,p) + C_2 W_2(z,p), \tag{40}$$

and we take $W_0(z,p) = W_1(z,p) = W(pz)$ and $W_2(z,p) = W(-pz)$. From the homogeneous solution, we have explicitly $W(pz) = W_{L,T}(pz) = \sqrt{pz}K_{\alpha_{L,T}}(pz)$ for the longitudinal and transversal components. The relative phase between $C_1^{L,T}$ and $C_2^{L,T}$ is fixed by requiring $U(z \to 0, p)$ being finite: $C_2^{L,T} = -C_1^{L,T}e^{i\pi(\alpha_{L,T}-\frac{1}{2})}$ for the longitudinal and transversal components, respectively.

Next, the remaining overall constant $C_1^{L,T}$ is determined by integrating over $z$ in Eq. (35), leading to

$$\frac{d}{dz}G_0^{L,T}(z \to z'-0^+) - \frac{d}{dz}G_0^{L,T}(z \to z'+0^+) = 1. \tag{41}$$

Plugging the general solution Eq. (39) into Eq. (41), for the longitudinal mode $G_L$ we have

$$1 = -C_1^L p e^{-i\frac{n\pi}{1-n}}\frac{ix}{2}\Big[K_{\alpha_L}(x)(K_{\alpha_L-1}(-x)+K_{\alpha_L+1}(-x)) + K_{\alpha_L}(-x)(K_{\alpha_L-1}(x)+K_{\alpha_L+1}(x))\Big], \tag{42}$$

where $x = pz'$. The right-hand side can be simplified using the properties of special functions (detailed in Appendix A.1), yielding the final result

$$1 = -\pi C_1^L p e^{-i\frac{n\pi}{1-n}}, \qquad C_1^L = \frac{1}{\pi p}e^{i\frac{\pi}{1-n}}. \tag{43}$$

Similarly, for the transverse mode $G_T$, we obtain

$$C_1^T = \frac{1}{\pi p}e^{i\frac{n\pi}{1-n}}. \tag{44}$$

With integral constants $C_{1,2}^{L,T}$, Eq. (40) can be simplified as $(x = pz)$[3]

$$\begin{aligned} \pi p \cdot U_L &= e^{i\frac{\pi}{1-n}}(W_L(x) - W_L(e^{i\pi}x)e^{-i\frac{n\pi}{1-n}}) = \pi\sqrt{x}I_{\alpha_L}(x), \\ \pi p \cdot U_T &= e^{i\frac{n\pi}{1-n}}(W_T(x) - W_T(e^{i\pi}x)e^{-i\frac{\pi}{1-n}}) = \pi\sqrt{x}I_{\alpha_T}(x). \end{aligned} \tag{45}$$

Finally, the Green's function in the mixed representation is given in terms of the modified Bessel function $K_\nu(x)$ and $I_\nu(x)$:[4]

$$G_0^{L,T}(p,z,z') = \begin{cases} \sqrt{zz'}I_{\alpha_{L,T}}(pz')K_{\alpha_{L,T}}(pz), & z > z', \\ \sqrt{zz'}K_{\alpha_{L,T}}(pz')I_{\alpha_{L,T}}(pz), & z < z'. \end{cases} \tag{46}$$

---

[2]For $n = 2$, the solution has elementary expressions

$$W_L(x) = e^{-x}\left(1 + \frac{3}{x} + \frac{3}{x^2}\right), \qquad W_T(x) = e^{-x}\left(1 + \frac{1}{x}\right), \tag{38}$$

for the longitudinal and transverse fields [22,35]. The derivation also appeared in Refs. [28,30].

[3]Here we use the property of $K_\nu(x)$: $K_\nu(e^{im\pi}x) = e^{-im\nu\pi}K_\nu(x) - i\pi\frac{\sin m\nu\pi}{\sin\nu\pi}I_\nu(x) \xrightarrow{m=1} e^{-i\nu\pi}K_\nu(x) - i\pi I_\nu(x)$.

[4]A similar expression has been shown in Ref. [9].

To derive the boundary operator expansion, we transform the Green's function to the layer susceptibility $\chi(z,z')$, following Ref. [26,27]. Assuming $z < z'$, we have

$$\chi_0^{L,T}(z,z') = \sqrt{zz'} \lim_{p \to 0} K_{\alpha_{L,T}}(pz')I_{\alpha_{L,T}}(pz) = \sqrt{zz'}\frac{1}{2\alpha_{L,T}}\left(\frac{z}{z'}\right)^{\alpha_{L,T}}, \qquad z < z'. \tag{47}$$

For $z > z'$, we just need to exchange $z$ and $z'$. Hence, with the variable $\zeta = \frac{\min(z,z')}{\max(z,z')}$, the mean-field layer susceptibility can be compactly expressed by

$$\chi_0^{L,T}(z,z') = \sqrt{zz'}\frac{\zeta^{\alpha_{L,T}}}{2\alpha_{L,T}}. \tag{48}$$

In mean-field theory, the scaling dimension of the $O(N)$ field is $\Delta_\phi = \frac{d-2}{2}$. Comparing this with the BOE in Eq. (27), we have $\Delta = \frac{d-1}{2} + \alpha_{L,T}$ and $c_\Delta = \frac{1}{4\alpha_{L,T}}$, which means that the corresponding longitudinal and transverse boundary modes have the scaling dimensions $\Delta_{L,T} = \frac{d-1}{2} + \alpha_{L,T}$, respectively.

In the following, we focus on the tricritical $O(N)$ model with $n = 3$. Substituting $n = 3$ and $c_0 \to -\infty$ into Eq. (31), the mean-field order parameter profile is

$$m_0(z) = \left(\frac{90}{u_0}\right)^{1/4} z^{-1/2}, \tag{49}$$

and the layer susceptibility is

$$\chi_0^L(z,z') = \sqrt{4zz'}\frac{\zeta^2}{8}, \qquad \chi_0^T(z,z') = \sqrt{4zz'}\frac{\zeta}{4}, \tag{50}$$

where we have used $\alpha_T = \alpha_L - 1 = 1$ for $n = 3$. Finally, from the BOE, we can also get the mean-field Green's function according to the relation Eq. (23). With $\Delta_T = \Delta_L - 1 = \frac{d+1}{2}$, the mean-field Green's function is given by

$$
\begin{aligned}
G_0^L(r,z,z') &= (4zz')^{-\frac{d-2}{2}}\frac{1}{8}\frac{\Gamma(\frac{d+3}{2})}{16\pi^{\frac{d-1}{2}}}\xi^{-\frac{d+3}{2}} \, {}_2F_1\left(\frac{d+3}{2},\frac{5}{2},5;-\xi^{-1}\right), \\
G_0^T(r,z,z') &= (4zz')^{-\frac{d-2}{2}}\frac{1}{4}\frac{\Gamma(\frac{d+1}{2})}{4\pi^{\frac{d-1}{2}}}\xi^{-\frac{d+1}{2}} \, {}_2F_1\left(\frac{d+1}{2},\frac{3}{2},3;-\xi^{-1}\right).
\end{aligned}
\tag{51}
$$

Notice that, here, we do not take $d = 3$ directly as we will implement the dimension regularization by setting $d = 3 - \epsilon$ in the next section.

## 3  Order parameter and layer susceptibility

In this section, we calculate the order parameter profile and the correlation functions for the longitudinal and transverse modes up to one-loop order. Readers who are not interested in the technical details of the loop calculations may skip this section.

At first, we briefly review the RG flow for the tricritical $O(N)$ model [22]. The bare coupling $u_0$ and the renormalized coupling $u$ are related via the renormalization factor $Z_u$, which at one-loop order is given by

$$u_0 = Z_u u = \left(1 + \frac{1}{2\pi^2}\frac{3N+22}{240}\frac{u}{\epsilon}\right)u, \tag{52}$$

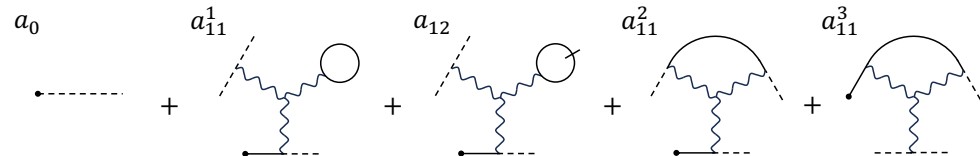

Figure 1: The Feynman diagrams for the order parameter profile $m(z) = \langle \phi_1(r, z) \rangle$. We use the dash line, wave line, and black dot to label $m_0 = \langle \phi_1(r, z) \rangle_0$, the $\phi^6$ vertex and $\phi_1(r, z)$, respectively. The solid line with (without) the short vertical tick corresponds to the transverse (longitudinal) Green's function.

where $d = 3 - \epsilon$. This leads to the following RG equation,

$$\frac{\mathrm{d}u}{\mathrm{d}\log\mu} = \beta_u, \qquad \beta_u = -\epsilon u + \frac{1}{2\pi^2}\frac{3N+22}{240}u^2. \tag{53}$$

According to the RG equation, the theory exhibits a nontrivial IR fixed point, known as the tricritical $O(N)$ fixed point,

$$u^* = \frac{240}{3N+22} \cdot 2\pi^2\epsilon, \tag{54}$$

while the field renormalization factor remains trivial at one-loop order, i.e., $Z_\phi = 1$.

For $\epsilon = 0$, the RG equation yields $u \sim \frac{1}{\log\mu_0 z}$ at large $z$, where $\mu_0$ is an arbitrary momentum scale, and $z$ serves as a length scale. Therefore, it corresponds to a logarithmic IR flow toward the Gaussian fixed point, as expected at the upper critical dimension $d = 3$ for the tricritical theory. It is important to keep this logarithmic flow instead of directly setting $\epsilon = 0$ in the fixed-point value of $u^*$. This was also emphasized in Ref. [26] in the context of the $\phi^4$ theory.

## 3.1 Order parameter profile

We consider the order parameter profile, $m(z) = \langle \phi_1(r, z) \rangle$, at one-loop order. The relevant Feynman diagram is shown in Fig. 1. We label each diagram as $a_{ij}$, where $i$ represents the number of vertices. For example, $a_{1j}$ ($a_{2j}$) corresponds to a Feynman diagram with one (two) vertex (vertices). The subscript $j$ refers to the loop formed by either the longitudinal or transversal propagators, as shown in Fig. 1. The contributions to $a_{ij}$ can be further categorized into different subdiagrams, denoted by the superscript $a_{ij}^k$. It is clear from Fig. 1, and $a_{ij} = \sum_k a_{ij}^k$.

The zeroth-order and first-order contributions to $m(z)$ are denoted as $m_0(z)$ and $m_1(z)$, respectively. The symmetry factors for the one-vertex diagrams in the first line of Fig. 1 are 12, $12(N-1)$, 24 and 24 for $a_{11}^1$, $a_{12}$, $a_{11}^2$ and $a_{11}^3$, respectively. Therefore, the order parameter profile can be written as $m(z) = m_0(z) + m_1(z)$, where $m_0(z) = a_0$, $m_1(z) = 60a_{11} + 12(N-1)a_{12}$, and $a_{11}$ and $a_{12}$ are given by

$$a_{11} = \int \mathrm{d}^{d-1}r\mathrm{d}y \frac{-u_0}{6!}G_0^L(r_0 - r, z, y)(m_0(y))^3 G_0^L(r = 0, y, y),$$
$$\tag{55}$$
$$a_{12} = \int \mathrm{d}^{d-1}r\mathrm{d}y \frac{-u_0}{6!}G_0^L(r_0 - r, z, y)(m_0(y))^3 G_0^T(r = 0, y, y),$$

with $G_0^{L,T}(r, z, z')$ being the propagator in Eq. (46). They can be further expressed in terms of the layer susceptibility as

$$a_{11} = \int dy \frac{-u_0}{6!} \chi_0^L(z, y)(m_0(y))^3 G_0^L(r = 0, y, y),$$

$$a_{12} = \int dy \frac{-u_0}{6!} \chi_0^L(z, y)(m_0(y))^3 G_0^T(r = 0, y, y). \tag{56}$$

To proceed, recall that $\xi = \frac{(r-r')^2 + (z-z')^2}{4zz'}$, then from the Green's function Eq. (51), we have

$$G_0^L(r = 0, y, y) = \frac{d+1}{5-d} \frac{\Gamma(\frac{d+1}{2})\Gamma(\frac{2-d}{2})}{2^d \pi^{\frac{d}{2}} \Gamma(\frac{5-d}{2})} y^{2-d} \equiv \gamma_L y^{2-d},$$

$$G_0^T(r = 0, y, y) = \frac{\Gamma(\frac{d+1}{2})\Gamma(\frac{2-d}{2})}{2^d \pi^{\frac{d}{2}} \Gamma(\frac{5-d}{2})} y^{2-d} \equiv \gamma_T y^{2-d}. \tag{57}$$

Substituting this to $a_{11}$ and performing the integration, we obtain

$$a_{11} = \frac{-u_0}{6!} \left(\frac{90}{u_0}\right)^{\frac{3}{4}} \frac{\gamma_L}{(4-d)d} z^{\frac{5}{2}-d}, \tag{58}$$

and similarly, $a_{12}$ is given by replacing $\gamma_L$ by $\gamma_T$. Finally, $m_1(z) = 60a_{11} + 12(N-1)a_{12}$ is obtained explicitly as,

$$m_1(z) = \frac{-u_0}{6!} \left(\frac{90}{u_0}\right)^{\frac{3}{4}} \frac{60\gamma_L + 12(N-1)\gamma_T}{(4-d)d} z^{\frac{5}{2}-d} = \frac{-u_0}{12} \left(\frac{90}{u_0}\right)^{\frac{3}{4}} \frac{\frac{d+1}{5-d} + \frac{N-1}{5}}{(4-d)d} \gamma_T z^{\frac{5}{2}-d}. \tag{59}$$

Hence, to first order, the order parameter profile is

$$m(z) = m_0(z) + m_1(z) = \left(\frac{90}{u_0}\right)^{\frac{1}{4}} z^{-\frac{1}{2}} \left[1 - \sqrt{\frac{5u_0}{8} \frac{\frac{d+1}{5-d} + \frac{N-1}{5}}{(4-d)d}} \gamma_T z^{3-d}\right]. \tag{60}$$

Note that in the calculation, we have not considered the renormalization effect for the vertex itself. To incorporate the one-loop correction to the vertex, we substitute $u_0$ in Eq. (52) into Eq. (60), yielding

$$m(z) = \left(\frac{3(3N+22)}{16\pi^2}\right)^{\frac{1}{4}} \epsilon^{-\frac{1}{4}} z^{-\frac{1}{2}} \left\{2^{-\frac{1}{4}} + 2^{-\frac{3}{4}} \frac{9+N}{\sqrt{3(3N+22)}} z^{\epsilon} \epsilon^{\frac{1}{2}} + \dots\right\}, \tag{61}$$

where we have taken the fixed point value $u_0 = 2u^* = 4\pi^2 \frac{240}{3N+22} \epsilon$. Denoting the normalization of the one-point function by $a_\sigma$, namely, $m(z) = \frac{a_\sigma}{(2z)^{\Delta_\phi}}$, we have

$$a_\sigma \approx \left(\frac{3(3N+22)}{8\pi^2}\right)^{\frac{1}{4}} \epsilon^{-\frac{1}{4}} + \frac{9+N}{2} \left(\frac{1}{6(3N+22)\pi^2}\right)^{\frac{1}{4}} \epsilon^{\frac{1}{4}}, \tag{62}$$

where we have expanded the results w.r.t. $\epsilon$.

There are a few remarks. (i) In principle, one should also include the field renormalization factor, $Z_\phi$, in the expression $m(z) = \langle\phi\rangle = \langle\phi_{\text{bare}}\rangle/\sqrt{Z_\phi}$, but here we have $Z_\phi = 1$ [22] in the one-loop order. (ii) The leading term in Eq. (62) comes from $m_0(z)$ with $u_0 = 2u^*$. (iii) Strictly at $d = 3$, a naive substitution of $\epsilon = 0$ in $a_\sigma$ leads to an unphysical divergence in the order parameter. Instead, we should consider the logarithmic flow of the coupling in Eq. (60), which gives $m(z) \sim \frac{(\log \mu_0 z)^{1/4}}{z^{1/2}}$. The order parameter vanishes deep in the bulk $z \to \infty$ and receives a logarithmic correction, as expected at the upper critical dimension.

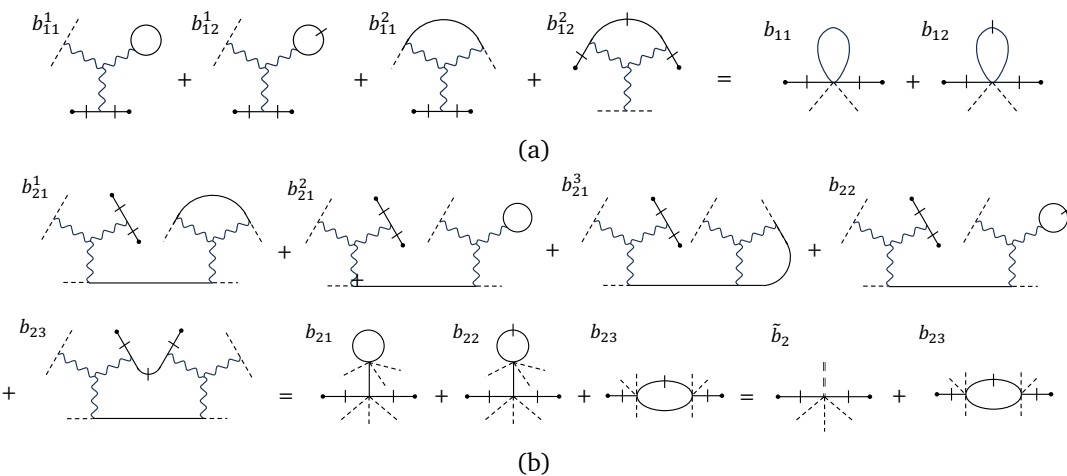

(a)

(b)

Figure 2: The Feynman diagrams for the Green's function of the transverse mode $G^T(r-r',z,z') = \langle \phi_i(r,z)\phi_i(r',z') \rangle$. $b_{ij} = \sum_k b_{ij}^k$. (a) The Feynman diagrams with one vertex. (b) The Feynman diagrams with two vertices. Here we combine $b_{21}$ and $b_{22}$, and express it with $\tilde{b}_2$.

## 3.2 Layer susceptibility of the transverse mode

We analyze the two-point function of the transverse mode $\langle \phi_i(z)\phi_i(z') \rangle$ with $i \neq 1$ at one-loop order. The corresponding Feynman diagrams are depicted in Fig 2. We introduce the notation $b_{ij}^k$ to denote the diagram with $i$ vertices. The symmetry factor for $b_{11}^1$, $b_{12}^1$, $b_{11}^2$, and $b_{12}^2$ in Fig. 2 (a) are 12, $12(N-1)$, 24 and 12, respectively. The total contribution from single-vertex diagrams is $36 b_{11} + 12 N b_{12}$, where $b_{11}$ and $b_{12}$ are shown on the right-hand side of Fig. 2 (a) and are given by

$$
\begin{aligned}
b_{11} &= \frac{-u_0}{6!} \int d^{d-1} r \, dy (m_0(y))^2 G_0^L(0,y,y) G_0^T(r_1 - r, z_1, y) G_0^T(r_2 - r, z_2, y), \\
b_{12} &= \frac{-u_0}{6!} \int d^{d-1} r \, dy (m_0(y))^2 G_0^T(0,y,y) G_0^T(r_1 - r, z_1, y) G_0^T(r_2 - r, z_2, y).
\end{aligned}
\tag{63}
$$

Here and after, we transform the Green's function into the layer susceptibility to simplify the calculations. The Green's function can later be reconstructed from the layer susceptibility using the BOE formalism detailed in the next section. The corresponding layer susceptibility is denoted with $(\cdot)'$:

$$
\begin{aligned}
b_{11}' &= \int d^{d-1} r_2 \, b_{11} = \frac{-u_0}{6!} \int dy (m_0(y))^2 G_0^L(0,y,y) \chi_0^T(z_1,y) \chi_0^T(z_2,y), \\
b_{12}' &= \int d^{d-1} r_2 \, b_{12} = \frac{-u_0}{6!} \int dy (m_0(y))^2 G_0^T(0,y,y) \chi_0^T(z_1,y) \chi_0^T(z_2,y).
\end{aligned}
\tag{64}
$$

Recall that the bare propagator and layer susceptibility are given in Eq. (51) and Eq. (50). We can perform the integration over $y$:

$$
\begin{aligned}
b_{11}' &= \frac{-u}{6!} \int dy \left(\frac{90}{u}\right)^{\frac{1}{2}} y^{-1} \gamma_L y^{2-d} \sqrt{4 z_1 y} \frac{1}{4} \frac{\min(z_1,y)}{\max(z_1,y)} \sqrt{4 z_2 y} \frac{1}{4} \frac{\min(z_2,y)}{\max(z_2,y)} \\
&= \frac{\gamma_L \sqrt{\frac{u}{10}}}{48(d-5)(d-3)(d-1)} (z_1 z_2)^{\frac{4-d}{2}} \left((d-1)\zeta^{\frac{5-d}{2}} + (d-5)\zeta^{\frac{d-1}{2}}\right),
\end{aligned}
\tag{65}
$$

where $\zeta = \frac{\min(z_1,z_2)}{\max(z_1,z_2)}$. Similarly, $b'_{12}$ is obtained by replacing $\gamma_L$ with $\gamma_T$.

Next, we calculate the two-vertex diagrams shown in Fig. 2 (b), denoted by $b^1_{21}$, $b^2_{21}$, $b^3_{21}$, $b_{22}$ and $b_{23}$. The symmetry factors are 576, 288, 576, 288$(N-1)$ and 576, respectively. Combining them, as shown in the middle expression in Fig. 2 (b), the two-vertex diagram can be expressed as $1440b_{21} + 288(N-1)b_{22} + 576b_{23}$. Further, comparing $1440b_{21} + 288(N-1)b_{22}$ with $m_1$ in Fig. 1, we find that the loop in $1440b_{21} + 288(N-1)b_{22}$ can be expressed by $m_1(z)$. Effectively, $1440b_{21} + 288(N-1)b_{22} = 24\tilde{b}_2$, where we used $\tilde{b}_2 = 60b_{21} + 12(N-1)b_{22}$ to indicate the first diagram in the last expression in Fig. 2 (b). After a straightforward integration, the corresponding layer susceptibility is

$$
\begin{aligned}
\tilde{b}'_2 &= \frac{-u_0}{6!} \int dy (m_0(y))^3 \chi_0^T(z_1,y)\chi_0^T(z_2,y)m_1(y) \\
&= \frac{\gamma_T\sqrt{\frac{5u_0}{2}}(\frac{d+1}{5-d}+\frac{N-1}{5})}{32(d-5)(d-4)(d-3)(d-1)d}(z_1z_2)^{\frac{4-d}{2}}\left((d-1)\zeta^{\frac{5-d}{2}}+(d-5)\zeta^{\frac{d-1}{2}}\right).
\end{aligned}
\tag{66}
$$

Now, let's focus on the calculation of $b_{23}$ shown in the last diagram in Fig. 2 (b). The corresponding layer susceptibility is given by

$$
b'_{23} = \left(\frac{-u_0}{6!}\right)^2 \int dy_1 dy_2 \chi_0^T(z_1,y_1)\chi_0^T(z_2,y_2)(m_0(y_1))^3(m_0(y_2))^3 B^{LT}(y_1,y_2),
\tag{67}
$$

where $B^{LT}(y_1,y_2) = \int d^{d-1}r\, G_0^T(r,y_1,y_2)G_0^L(r,y_1,y_2)$. We first calculate $B^{LT}(y_1,y_2)$. After Fourier transformation in the hypersurface parallel to the boundary, we can use the mixed representation of Green's functions to simplify

$$
\begin{aligned}
B^{LT}(y_1,y_2) &= \int \frac{d^{d-1}\vec{p}}{(2\pi)^{d-1}} G_0^T(\vec{p},y_1,y_2)G_0^L(-\vec{p},y_1,y_2) \\
&= \frac{S_{d-1}}{(2\pi)^{d-1}} \int dp\, p^{d-2} G_0^T(p,y_1,y_2)G_0^L(p,y_1,y_2) \\
&= \frac{S_{d-1}}{(2\pi)^{d-1}} y_1 y_2 \int dp\, p^{d-2} I_1(py_1)K_1(py_2)I_2(py_1)K_2(py_2),
\end{aligned}
\tag{68}
$$

where $S_d = \frac{2\pi^{d/2}}{\Gamma(d/2)}$ is the area of a unit sphere in $d$ dimensions. In going from the second line to the last line, we used the bare propagator in Eq. (46) and assumed $y_1 < y_2$. The details of evaluation for the integral $\int dp\, p^{d-2} I_1(py_1)K_1(py_2)I_2(py_1)K_2(py_2)$ in the last line can be found in Appendix B.1. Taking $v = 1$ in Eq. (B.2), we obtain

$$
\begin{aligned}
B^{LT}(y_1,y_2) &= \frac{S_{d-1}}{(2\pi)^{d-1}}\frac{1}{4}y_1^{3-d}\left(\frac{y_1}{y_2}\right)^{d+1}\Gamma\left(\frac{d-1}{2}\right)\Gamma\left(\frac{5}{2}\right)\Gamma\left(\frac{d+1}{2}\right)\Gamma\left(\frac{d+5}{2}\right) \\
&\quad \times {}_4\tilde{F}_3\left[\left\{\frac{d-1}{2},\frac{5}{2},\frac{d+1}{2},\frac{d+5}{2}\right\},\left\{3,\frac{d+2}{2},4\right\},\left(\frac{y_1}{y_2}\right)^2\right],
\end{aligned}
\tag{69}
$$

in which ${}_p\tilde{F}_q$ is the regularized generalized hypergeometric function ${}_p\tilde{F}_q[a,b,z] = \frac{{}_pF_q[a,b,z]}{\Gamma(b_1)\cdots\Gamma(b_q)}$ with $a = \{a_1,\cdots,a_p\}$ and $b = \{b_1,\cdots,b_q\}$. For $y_1 > y_2$ we just exchange $y_1$ and $y_2$ in Eq. (69). With this result, $b'_{23}$ can be simplified as follows,

$$
\begin{aligned}
b'_{23} &= \left(\frac{-u_0}{6!}\right)^2\left(\frac{90}{u}\right)^{\frac{3}{2}}\frac{1}{4}\sqrt{z_1z_2}\left(\int_0^\infty dy_1\int_{y_1}^\infty dy_2 + \int_0^\infty dy_2\int_{y_2}^\infty dy_1\right)\frac{1}{y_1y_2} \\
&\quad \times \frac{\min(z_1,y_1)}{\max(z_1,y_1)}\frac{\min(z_2,y_2)}{\max(z_2,y_2)}B^{LT}(\min(y_1,y_2),\max(y_1,y_2)) \\
&\equiv \left(\frac{-u_0}{6!}\right)^2\left(\frac{90}{u}\right)^{\frac{3}{2}}\frac{1}{4}\sqrt{z_1z_2}\frac{S_{d-1}}{(2\pi)^{d-1}}\frac{1}{4}\Gamma\left(\frac{d-1}{2}\right)\Gamma\left(\frac{5}{2}\right)\Gamma\left(\frac{d+1}{2}\right)\Gamma\left(\frac{d+5}{2}\right)(I_1+I_2),
\end{aligned}
\tag{70}
$$

where $I_1$ and $I_2$ denote two integrals whose evaluations are detailed in Appendix B.2. Using the results in Eq. (B.7), $b'_{23}$ is

$$b'_{23} = \frac{90^{\frac{3}{2}}}{(6!)^2} \frac{S_{d-1}}{32(2\pi)^{d-1}} \sqrt{u_0}(z_1 z_2)^{\frac{4-d}{2}} \left( A C_1 \zeta^{\frac{5-d}{2}} + B C_2 \zeta^{\frac{d-1}{2}} \right) + (z_1 z_2)^{\frac{4-d}{2}} H(\zeta), \tag{71}$$

where the constants $C_1$ and $C_2$ are given explicitly in Appendix B.2. Here we show the numerical values for these two constants in $\mathcal{O}(\epsilon)$,

$$C_1 = \frac{4}{3} + c_{1,1}\epsilon + \mathcal{O}(\epsilon^2), \qquad C_2 = \frac{4}{3} + c_{2,1}\epsilon + \mathcal{O}(\epsilon^2), \tag{72}$$

where $c_{1,1} \approx -1.39608$ and $c_{2,1} \approx -1.48791$.

The last term in $b'_{23}$ is explicitly given by

$$
\begin{aligned}
H(\zeta) = {} & \frac{90^{\frac{3}{2}}}{(6!)^2} \frac{S_{d-1}}{32(2\pi)^{d-1}} \Gamma\left(\frac{d-1}{2}\right)\Gamma\left(\frac{5}{2}\right)\Gamma\left(\frac{d+1}{2}\right)\Gamma\left(\frac{d+5}{2}\right) \sqrt{u_0}\, \zeta^{\frac{d+5}{2}} \\
& \times \Bigg\{ A\Gamma\left(\frac{5}{2}\right) {}_5\tilde{F}_4\left[ \left\{ \frac{5}{2}, \frac{d-1}{2}, \frac{5}{2}, \frac{d+1}{2}, \frac{d+5}{2} \right\}, \left\{ \frac{7}{2}, 3, \frac{d+2}{2}, 4 \right\}, \zeta^2 \right] \\
& \quad - A\Gamma\left(\frac{d}{2}\right) {}_5\tilde{F}_4\left[ \left\{ \frac{d}{2}, \frac{d-1}{2}, \frac{5}{2}, \frac{d+1}{2}, \frac{d+5}{2} \right\}, \left\{ \frac{d+2}{2}, 3, \frac{d+2}{2}, 4 \right\}, \zeta^2 \right] \\
& \quad + B\Gamma\left(\frac{d+2}{2}\right) {}_5\tilde{F}_4\left[ \left\{ \frac{d+2}{2}, \frac{d-1}{2}, \frac{5}{2}, \frac{d+1}{2}, \frac{d+5}{2} \right\}, \left\{ \frac{d+4}{2}, 3, \frac{d+2}{2}, 4 \right\}, \zeta^2 \right] \\
& \quad - B\Gamma\left(\frac{3}{2}\right) {}_5\tilde{F}_4\left[ \left\{ \frac{3}{2}, \frac{d-1}{2}, \frac{5}{2}, \frac{d+1}{2}, \frac{d+5}{2} \right\}, \left\{ \frac{5}{2}, 3, \frac{d+2}{2}, 4 \right\}, \zeta^2 \right] \Bigg\},
\end{aligned} \tag{73}
$$

with $A = \frac{-2}{(d-5)(d-3)}$, $B = \frac{-2}{(d-3)(d-1)}$. It is obvious that for $d = 3 - \epsilon$, the first two terms in Eq. (71) are the leading contribution, and $H(\zeta)$ corresponds to higher-order contributions.

Now, adding all contributions together, the layer susceptibility of the transverse mode is

$$
\begin{aligned}
\chi^T(z_1, z_2) = {} & \chi_0^T(z_1, z_2) + 36 b'_{11} + 12N b'_{12} + 24\tilde{b}'_2 + 576 b'_{23} \\
= {} & \frac{1}{2}\sqrt{z_1 z_2}\, \zeta - \left( \frac{3(d+1)}{5-d} + N + \frac{15\left(\frac{d+1}{5-d} + \frac{N-1}{5}\right)}{(d-4)d} \right) \frac{\gamma_T}{8} \sqrt{\frac{u_0}{10}} (z_1 z_2)^{\frac{4-d}{2}} \left( A\zeta^{\frac{5-d}{2}} + B\zeta^{\frac{d-1}{2}} \right) \\
& + \frac{3}{32} \frac{S_{d-1}}{(2\pi)^{d-1}} \sqrt{\frac{u_0}{10}} (z_1 z_2)^{\frac{4-d}{2}} \left( A C_1 \zeta^{\frac{5-d}{2}} + B C_2 \zeta^{\frac{d-1}{2}} \right) + 576 (z_1 z_2)^{\frac{4-d}{2}} H(\zeta).
\end{aligned} \tag{74}
$$

## 3.3 Layer susceptibility of the longitudinal mode

We consider the connected two-point functions of the longitudinal mode $\langle \phi_1(z)\phi_1(z') \rangle_{\text{con}}$ with the one-loop Feynman diagrams shown in Fig 3. Using the same method as in Sec. 3.2, we can evaluate the diagrams straightforwardly. The symmetry factors of $c_{11}$ and $c_{12}$ in Fig. 3 (a) are 180 and $36(N-1)$, respectively. We now directly compute the corresponding layer susceptibility

$$
\begin{aligned}
c'_{11} &= \frac{-u_0}{6!} \int dy (m_0(y))^2 G_0^L(0, y, y) \chi_0^L(z_1, y) \chi_0^L(z_2, y) \\
&= -\frac{\gamma_L \sqrt{\frac{u_0}{10}}}{384} (z_1 z_2)^{\frac{4-d}{2}} \left( A'\zeta^{\frac{d+1}{2}} + B'\zeta^{\frac{7-d}{2}} \right), \\
c'_{12} &= \frac{-u_0}{6!} \int dy (m_0(y))^2 G_0^T(0, y, y) \chi_0^L(z_1, y) \chi_0^L(z_2, y) \\
&= -\frac{\gamma_T \sqrt{\frac{u_0}{10}}}{384} (z_1 z_2)^{\frac{4-d}{2}} \left( A'\zeta^{\frac{d+1}{2}} + B'\zeta^{\frac{7-d}{2}} \right),
\end{aligned} \tag{75}
$$

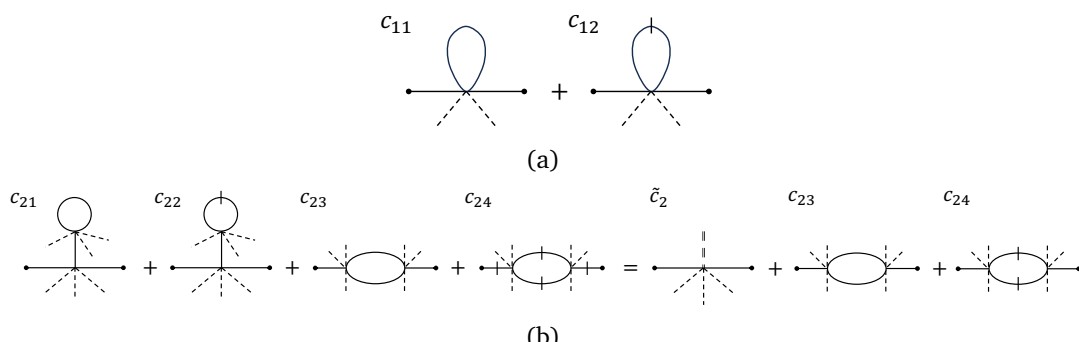

Figure 3: The Feynman diagram for the connected Green's function of the longitudinal mode $G^L(r-r',z,z')$. (a) The Feynman diagrams with one vertex. (b) The Feynman diagrams with two vertices. Here we combine $c_{21}$ and $c_{22}$, and express it with $\tilde{c}_2$.

where we used the mean-field functions in Eqs. (49), (51), (50) and performed the integration. Here $\zeta = \frac{\min(z_1,z_2)}{\max(z_1,z_2)}$, $A' = \frac{-4}{(d-3)(d+1)}$ and $B' = \frac{-4}{(d-7)(d-3)}$.

Next, we calculate the two-vertex diagrams shown in Fig. 3 (b). The combination of $c_{21}$ and $c_{22}$ can be expressed again with the help of $m_1$ in Fig. 1. Hence, we define this combination as $\tilde{c}_2$. The symmetry factors are 120, 7776 and $288(N-1)$ for $\tilde{c}_2$, $c_{23}$ and $c_{24}$, respectively, leading to the two-vertex correction: $120\tilde{c}_2 + 7776c_{23} + 288(N-1)c_{24}$. We evaluate the layer susceptibility corresponding to $\tilde{c}_2$, $c_{23}$ and $c_{24}$ in the following.

The calculation for $\tilde{c}_2$ is straightforward:

$$
\begin{aligned}
\tilde{c}_2' &= \frac{-u_0}{6!} \int \mathrm{d}y (m_0(y))^3 \chi_0^L(z_1,y) \chi_0^L(z_2,y) m_1(y) \\
&= -\frac{\gamma_T \sqrt{\frac{5u_0}{2}} \left(\frac{d+1}{5-d} + \frac{N-1}{5}\right)}{256(d-4)d} (z_1 z_2)^{\frac{4-d}{2}} \left(A'\zeta^{\frac{d+1}{2}} + B'\zeta^{\frac{7-d}{2}}\right),
\end{aligned}
\tag{76}
$$

where to get the second line, an integration over $y$ has been performed.

The evaluations for diagrams $c_{23}$ and $c_{24}$ are similar, so we present them together. The corresponding layer susceptibilities are

$$
\begin{aligned}
c_{23}' &= \left(\frac{-u_0}{6!}\right)^2 \int \mathrm{d}y_1 \mathrm{d}y_2 \chi_0^L(z_1,y_1) \chi_0^L(z_2,y_2) (m_0(y_1))^3 (m_0(y_2))^3 B^{LL}(y_1,y_2), \\
c_{24}' &= \left(\frac{-u_0}{6!}\right)^2 \int \mathrm{d}y_1 \mathrm{d}y_2 \chi_0^L(z_1,y_1) \chi_0^L(z_2,y_2) (m_0(y_1))^3 (m_0(y_2))^3 B^{TT}(y_1,y_2),
\end{aligned}
\tag{77}
$$

where $B^{LL}(y_1,y_2) = \int \mathrm{d}^{d-1}r (G_0^L(r,y_1,y_2))^2$ and $B^{TT}(y_1,y_2) = \int \mathrm{d}^{d-1}r (G_0^T(r,y_1,y_2))^2$. Via a similar Fourier transformation in the parallel hyperspace, we arrive at

$$
\begin{aligned}
B^{LL}(y_1,y_2) &= \frac{S_{d-1}}{(2\pi)^{d-1}} y_1 y_2 \int \mathrm{d}p\, p^{d-2} I_2(py_1) K_2(py_2) I_2(py_1) K_2(py_2) \\
&= \frac{S_{d-1}}{(2\pi)^{d-1}} \frac{1}{4} y_1^{3-d} \left(\frac{y_1}{y_2}\right)^{2+d} \Gamma\left(\frac{5}{2}\right) \Gamma\left(\frac{d-1}{2}\right) \Gamma\left(\frac{d+3}{2}\right) \Gamma\left(\frac{d+7}{2}\right) \\
&\quad \times {}_4\tilde{F}_3\left[\left\{\frac{5}{2}, \frac{d-1}{2}, \frac{d+3}{2}, \frac{d+7}{2}\right\}, \left\{3, 5, \frac{d+4}{2}\right\}, \left(\frac{y_1}{y_2}\right)^2\right],
\end{aligned}
\tag{78}
$$

$$B^{TT}(y_1, y_2) = \frac{S_{d-1}}{(2\pi)^{d-1}} y_1 y_2 \int \mathrm{d}p\, p^{d-2} I_1(py_1) K_1(py_2) I_1(py_1) K_1(py_2)$$

$$= \frac{S_{d-1}}{(2\pi)^{d-1}} \frac{1}{4} y_1^{3-d} \left(\frac{y_1}{y_2}\right)^d \Gamma\left(\frac{3}{2}\right) \Gamma\left(\frac{d-1}{2}\right) \Gamma\left(\frac{d+1}{2}\right) \Gamma\left(\frac{d+3}{2}\right)$$

$$\times {}_4\tilde{F}_3\left[\left\{\frac{3}{2}, \frac{d-1}{2}, \frac{d+1}{2}, \frac{d+3}{2}\right\}, \left\{2, 3, \frac{d+2}{2}\right\}, \left(\frac{y_1}{y_2}\right)^2\right].$$

The details of evaluation for the integral $\int \mathrm{d}p\, p^{d-2} I_1(py_1) K_1(py_2) I_2(py_1) K_2(py_2)$ in the last line can be found in Appendix B.1. Here we have assumed $y_1 < y_2$, but the result for $y_1 > y_2$ can be obtained by exchanging $y_1$ and $y_2$. Hence, $c'_{23}$ and $c'_{24}$ can be evaluated as follows,

$$c'_{23} = \left(\frac{-u_0}{6!}\right)^2 \left(\frac{90}{u_0}\right)^{\frac{3}{2}} \frac{1}{16} \sqrt{z_1 z_2} \left(\int_0^\infty \mathrm{d}y_1 \int_{y_1}^\infty \mathrm{d}y_2 + \int_0^\infty \mathrm{d}y_2 \int_{y_2}^\infty \mathrm{d}y_1\right) \frac{1}{y_1 y_2} \tag{79}$$

$$\times \left(\frac{\min(z_1, y_1)}{\max(z_1, y_1)}\right)^2 \left(\frac{\min(z_2, y_2)}{\max(z_2, y_2)}\right)^2 B^{LL}(\min(y_1, y_2), \max(y_1, y_2))$$

$$\equiv \left(\frac{-u_0}{6!}\right)^2 \left(\frac{90}{u_0}\right)^{\frac{3}{2}} \frac{1}{16} \sqrt{z_1 z_2} \frac{S_{d-1}}{(2\pi)^{d-1}} \frac{1}{4} \Gamma\left(\frac{5}{2}\right) \Gamma\left(\frac{d-1}{2}\right) \Gamma\left(\frac{d+3}{2}\right) \Gamma\left(\frac{d+7}{2}\right) (I'_1 + I'_2),$$

$$c'_{24} = \left(\frac{-u_0}{6!}\right)^2 \left(\frac{90}{u_0}\right)^{\frac{3}{2}} \frac{1}{16} \sqrt{z_1 z_2} \left(\int_0^\infty \mathrm{d}y_1 \int_{y_1}^\infty \mathrm{d}y_2 + \int_0^\infty \mathrm{d}y_2 \int_{y_2}^\infty \mathrm{d}y_1\right) \frac{1}{y_1 y_2}$$

$$\times \left(\frac{\min(z_1, y_1)}{\max(z_1, y_1)}\right)^2 \left(\frac{\min(z_2, y_2)}{\max(z_2, y_2)}\right)^2 B^{TT}(\min(y_1, y_2), \max(y_1, y_2))$$

$$\equiv \left(\frac{-u_0}{6!}\right)^2 \left(\frac{90}{u_0}\right)^{\frac{3}{2}} \frac{1}{16} \sqrt{z_1 z_2} \frac{S_{d-1}}{(2\pi)^{d-1}} \frac{1}{4} \Gamma\left(\frac{3}{2}\right) \Gamma\left(\frac{d-1}{2}\right) \Gamma\left(\frac{d+1}{2}\right) \Gamma\left(\frac{d+3}{2}\right) (I''_1 + I''_2),$$

where $I'_{1,2}$ and $I''_{1,2}$ are integrals detailed in Appendix B.3. The final result for $c'_{23}$ is

$$c'_{23} = \frac{90^{\frac{3}{2}}}{(6!)^2} \frac{S_{d-1}}{128(2\pi)^{d-1}} \sqrt{u_0} (z_1 z_2)^{\frac{4-d}{2}} \left(A' C'_1 \zeta^{\frac{d+1}{2}} + B' C'_2 \zeta^{\frac{7-d}{2}}\right) + (z_1 z_2)^{\frac{4-d}{2}} H'(\zeta), \tag{80}$$

where $C'_1$ and $C'_2$ are given explicitly in Appendix B.3. Their numerical values up to $\mathcal{O}(\epsilon)$ are

$$C'_1 = \frac{16}{15} + c'_{1,1}\epsilon + \mathcal{O}(\epsilon^2), \qquad C'_2 = \frac{16}{15} + c'_{2,1}\epsilon + \mathcal{O}(\epsilon^2), \tag{81}$$

with $c'_{1,1} \approx -1.45699$ and $c'_{2,1} \approx -1.35775$. The last term in $c'_{23}$ is

$$H'(\zeta) = \frac{90^{\frac{3}{2}}}{(6!)^2} \frac{S_{d-1}}{128(2\pi)^{d-1}} \Gamma\left(\frac{5}{2}\right) \Gamma\left(\frac{d-1}{2}\right) \Gamma\left(\frac{d+3}{2}\right) \Gamma\left(\frac{d+7}{2}\right) \sqrt{u_0} \zeta^{\frac{d+7}{2}}$$

$$\times \left\{A' \Gamma\left(\frac{d+4}{2}\right) {}_5\tilde{F}_4\left[\left\{\frac{d+4}{2}, \frac{5}{2}, \frac{d-1}{2}, \frac{d+3}{2}, \frac{d+7}{2}\right\}, \left\{\frac{d+6}{2}, 3, 5, \frac{d+4}{2}\right\}, \zeta^2\right]\right.$$

$$-A' \Gamma\left(\frac{3}{2}\right) {}_5\tilde{F}_4\left[\left\{\frac{3}{2}, \frac{5}{2}, \frac{d-1}{2}, \frac{d+3}{2}, \frac{d+7}{2}\right\}, \left\{\frac{5}{2}, 3, 5, \frac{d+4}{2}\right\}, \zeta^2\right]$$

$$+B' \Gamma\left(\frac{7}{2}\right) {}_5\tilde{F}_4\left[\left\{\frac{7}{2}, \frac{5}{2}, \frac{d-1}{2}, \frac{d+3}{2}, \frac{d+7}{2}\right\}, \left\{\frac{9}{2}, 3, 5, \frac{d+4}{2}\right\}, \zeta^2\right]$$

$$\left.-B' \Gamma\left(\frac{d}{2}\right) {}_5\tilde{F}_4\left[\left\{\frac{d}{2}, \frac{5}{2}, \frac{d-1}{2}, \frac{d+3}{2}, \frac{d+7}{2}\right\}, \left\{\frac{d+2}{2}, 3, 5, \frac{d+4}{2}\right\}, \zeta^2\right]\right\}. \tag{82}$$

Similarly, $c'_{24}$ can be obtained as

$$c'_{24} = \frac{90^{\frac{3}{2}}}{(6!)^2} \frac{S_{d-1}}{128(2\pi)^{d-1}} \sqrt{u_0} (z_1 z_2)^{\frac{4-d}{2}} \left(A' C''_1 \zeta^{\frac{d+1}{2}} + B' C''_2 \zeta^{\frac{7-d}{2}}\right) + (z_1 z_2)^{\frac{4-d}{2}} H''(\zeta), \tag{83}$$

where

$$C_1'' = \frac{8}{3} + c_{1,1}''\epsilon + \mathcal{O}(\epsilon^2), \qquad C_2'' = \frac{8}{3} + c_{2,1}''\epsilon + \mathcal{O}(\epsilon^2),$$
$$c_{1,1}'' \approx -1.64248, \qquad\qquad c_{2,1}'' \approx -0.263237,$$

(84)

and

$$H''(\zeta) = \frac{90^{\frac{3}{2}}}{(6!)^2} \frac{S_{d-1}}{128(2\pi)^{d-1}} \Gamma\left(\frac{3}{2}\right) \Gamma\left(\frac{d-1}{2}\right) \Gamma\left(\frac{d+1}{2}\right) \Gamma\left(\frac{d+3}{2}\right) \sqrt{u_0} \zeta^{\frac{d+3}{2}}$$

$$\times \left\{ A'\Gamma\left(\frac{d+2}{2}\right) {}_5\tilde{F}_4\left[\left\{\frac{d+2}{2}, \frac{3}{2}, \frac{d-1}{2}, \frac{d+1}{2}, \frac{d+3}{2}\right\}, \left\{\frac{d+4}{2}, 2, 3, \frac{d+2}{2}\right\}, \zeta^2\right] \right.$$

$$- A'\Gamma\left(\frac{1}{2}\right) {}_5\tilde{F}_4\left[\left\{\frac{1}{2}, \frac{3}{2}, \frac{d-1}{2}, \frac{d+1}{2}, \frac{d+3}{2}\right\}, \left\{\frac{3}{2}, 2, 3, \frac{d+2}{2}\right\}, \zeta^2\right]$$

$$+ B'\Gamma\left(\frac{5}{2}\right) {}_5\tilde{F}_4\left[\left\{\frac{5}{2}, \frac{3}{2}, \frac{d-1}{2}, \frac{d+1}{2}, \frac{d+3}{2}\right\}, \left\{\frac{7}{2}, 2, 3, \frac{d+2}{2}\right\}, \zeta^2\right]$$

$$\left. - B'\Gamma\left(\frac{d-2}{2}\right) {}_5\tilde{F}_4\left[\left\{\frac{d-2}{2}, \frac{3}{2}, \frac{d-1}{2}, \frac{d+1}{2}, \frac{d+3}{2}\right\}, \left\{\frac{d}{2}, 2, 3, \frac{d+2}{2}\right\}, \zeta^2\right] \right\}.$$

(85)

The explicit expression for $C_{1,2}''$ can be found in Appendix B.3. It is obvious that for $d = 3-\epsilon$, the first two terms in $c_{23}'$ and $c_{24}'$ are the leading contribution, while $H'(\zeta)$ and $H''(\zeta)$ correspond to higher-order contributions.

In total, summing over all Feynman diagrams yields the layer susceptibility $\chi^L(z_1, z_2)$:

$$\chi^L(z_1, z_2) = \chi_0^L + 180c_{11}' + 36(N-1)c_{12}' + 120(c_{21} + c_{22})' + 7776c_{23}' + 288(N-1)c_{24}'$$

(86)

$$= \frac{1}{4}\sqrt{z_1 z_2}\zeta^2 - \left(\frac{15(d+1)}{5-d} + 3(N-1) + \frac{75\left(\frac{d+1}{5-d} + \frac{N-1}{5}\right)}{(d-4)d}\right) \frac{\gamma_T}{32}\sqrt{\frac{u_0}{10}}(z_1 z_2)^{\frac{4-d}{2}} \left(A'\zeta^{\frac{d+1}{2}} + B'\zeta^{\frac{7-d}{2}}\right)$$

$$+ \frac{3}{256}\frac{S_{d-1}}{(2\pi)^{d-1}}\sqrt{\frac{u_0}{10}}(z_1 z_2)^{\frac{4-d}{2}} \left(A'[27C_1' + (N-1)C_1'']\zeta^{\frac{d+1}{2}} + B'[27C_2' + (N-1)C_2'']\zeta^{\frac{7-d}{2}}\right)$$

$$+ 7776(z_1 z_2)^{\frac{4-d}{2}}H'(\zeta) + 288(N-1)(z_1 z_2)^{\frac{4-d}{2}}H''(\zeta).$$

## 4 Boundary operator expansion

Now, we are ready to consider the boundary operator expansion (BOE). Using the results in Section 3, the layer susceptibility can be expressed as a power series in $\zeta = \frac{\min(z,z')}{\max(z,z')}$. The corresponding prefactors yield the BOE coefficients in Eq. (23) and (27).

### 4.1 BOE of the transverse mode

As in Eq. (27), the layer susceptibility $\chi^T(z_1, z_2)$ can be expanded in the following form

$$\chi^T(z_1, z_2) = \sqrt{4z_1 z_2}\zeta^{-\frac{d-1}{2}}\left(c_{d-1}^T\zeta^{d-1} + \sum_{\Delta > d-1} c_\Delta^T \zeta^\Delta\right).$$

(87)

From the one-loop result in Eq. (74), the leading-order result gives

$$c_{d-1}^T = \frac{1}{4} - \frac{2 + 3c_{1,1} - 3c_{2,1}}{128\pi}\sqrt{\frac{u_0}{10}} = \frac{1}{4} - \frac{2 + 3c_{1,1} - 3c_{2,1}}{16}\sqrt{\frac{3}{2(3N+22)}}\epsilon^{\frac{1}{2}} + \mathcal{O}(\epsilon^{\frac{3}{2}}).$$

(88)

In the first equality, we set $d = 3$ while retaining the $u_0$ coupling. The appearance of seemingly $\mathcal{O}(\epsilon)$ contributions $c_{1,1}$ and $c_{2,1}$ is because of the leading $\mathcal{O}(1/\epsilon)$ contribution in the constants $A$ and $B$. In the last expression, we substitute $u_0 = Z_u u$ with the fixed-point value in Eq. (54). We expect that the appearance of the field with scaling dimension of $d-1$ at both the leading and subleading orders is related to the broken $O(N)$ symmetry at the boundary. It would be interesting to investigate the Ward identity associated with the $O(N)$ current and demonstrate this explicitly.

The higher-order terms in BOE come from $H(\zeta)$. We use the expansion of the regularized generalized hypergeometric function:

$$
{}_5\tilde{F}_4[\{a_1,\cdots,a_5\},\{b_1,\cdots,b_4\},z] = \sum_{k=0}^{\infty} \frac{\frac{\Gamma(a_1+k)}{\Gamma(a_1)}\frac{\Gamma(a_2+k)}{\Gamma(a_2)}\frac{\Gamma(a_3+k)}{\Gamma(a_3)}\frac{\Gamma(a_4+k)}{\Gamma(a_4)}\frac{\Gamma(a_5+k)}{\Gamma(a_5)}}{\Gamma(k+1)\Gamma(b_1+k)\Gamma(b_2+k)\Gamma(b_3+k)\Gamma(b_4+k)} z^k . \quad (89)
$$

With this expansion, $H(\zeta)$ can be rewritten as a series in $\zeta$,

$$
H(\zeta) = \frac{90^{\frac{3}{2}}}{(6!)^2} \frac{S_{d-1}}{32(2\pi)^{d-1}} \sqrt{u_0}\, \zeta^{\frac{d+5}{2}} \tag{90}
$$
$$
\times \sum_{k=0}^{\infty} \frac{\Gamma(\frac{d-1}{2}+k)\Gamma(\frac{5}{2}+k)\Gamma(\frac{d+1}{2}+k)\Gamma(\frac{d+5}{2}+k)}{\Gamma(k+1)\Gamma(3+k)\Gamma(\frac{d+2}{2}+k)\Gamma(4+k)} \left( \frac{A}{\frac{5}{2}+k} - \frac{A}{\frac{d}{2}+k} + \frac{B}{\frac{d+2}{2}+k} - \frac{B}{\frac{3}{2}+k} \right) \zeta^{2k} .
$$

Substituting this expansion to the layer susceptibility and comparing it with the coefficient defined in Eq. (87), we arrive at

$$
c_k^T = \frac{3}{8\pi} \frac{1}{k^2(k-1)(k-2)^2} \sqrt{\frac{u_0}{10}} = \frac{3}{2} \sqrt{\frac{6}{3N+22}} \frac{1}{k^2(k-1)(k-2)^2} \epsilon^{\frac{1}{2}}, \quad k = 5,7,9,\dots \quad (91)
$$

Note that in the first equality, we take $d = 3$ while keeping $u_0$. In the last expression, we further set $u_0$ to be the fixed-point value.

There are three additional remarks. (i) It is worth noting that, in contrast to the $\phi^4$ theory, the leading term in the $\epsilon$ expansion is $\mathcal{O}(\epsilon^{1/2})$. This is not a coincidence because the power of $\mathcal{O}(\epsilon^{1/2})$ also appears at the special transition [22]. (ii) Naively, setting $\epsilon = 0$ in the final expression of $c_k^T$ for $d = 3$ seems to indicate that the BOE vanishes at the leading order. However, in strictly $d = 3$, $u_0$ exhibits a logarithmic flow, $u_0 \sim \frac{1}{\log \mu_0 z}$. Substituting this logarithmic flow into the expression of the BOE with explicit $u_0$ dependence reveals a logarithmic contribution to the BOE, indicating that it is not entirely trivial. (iii) In general, with $c_k^T$ in Eq. (23), we can reconstruct the Green's function $G^T(\xi, z_1, z_2)$ by re-summing the infinite series. While this can be done explicitly for the critical $O(N)$ model with $n = 2$ [27], we are not able to get a compact form for the tricritical point. But we can get the expansion of $G^T(\xi, z_1, z_2)$ as a series in $\xi$, with known prefactors for each term.[5]

## 4.2 BOE of the longitudinal mode

The BOE of the layer susceptibility for the longitudinal mode can be obtained similarly. The layer susceptibility $\chi^L(z_1, z_2)$ is expanded in the following form,

$$
\chi^L(z_1, z_2) = \sqrt{4z_1 z_2}\, \zeta^{-\frac{d-1}{2}} \left( c_d^L \zeta^d + \sum_{\Delta > d} c_\Delta^L \zeta^\Delta \right) . \quad (92)
$$

We can simplify the first two lines in the last equality in Eq. (86), with all known constants $\gamma_T, A', B'$, and $C'_{1,2}, C''_{1,2}$. A further simplification is done by setting $d = 3$. The simplification

---

[5]We can express ${}_2\tilde{F}_1$ in the integration representation, expand with respect to $\xi$, and then implement the integration to get the prefactor of $\xi^n$.

is straightforward, so we skip the derivation. The final result gives the leading coefficient,

$$
\begin{aligned}
c_d^L &= \frac{1}{8} + \frac{16 + 405(c_{1,1}' - c_{2,1}') + 15(N-1)(c_{1,1}'' - c_{2,1}'')}{640} \frac{1}{8\pi} \sqrt{\frac{u_0}{10}} \\
&= \frac{1}{8} + \frac{16 + 405(c_{1,1}' - c_{2,1}') + 15(N-1)(c_{1,1}'' - c_{2,1}'')}{640} \sqrt{\frac{3}{2(3N+22)}} \epsilon^{\frac{1}{2}} + \mathcal{O}(\epsilon).
\end{aligned}
\tag{93}
$$

This operator with scaling dimension $d$ is the displacement operator. The presence of a boundary breaks the translation symmetry. As a result, the conservation of the stress-energy tensor is violated by a boundary term, the displacement operator. Hence, its scaling dimension is protected to be $d$.

$H'(\zeta)$ and $H''(\zeta)$ contribute to the higher-order coefficients. Using the same formula Eq. (89), and after some tedious manipulations not shown here, we obtain the higher-order terms in the expansion at the leading order in the $\epsilon$-expansion,

$$
\begin{aligned}
\sum_{\Delta=4}^{\infty} c_\Delta^L \zeta^\Delta = \frac{3}{1024\pi} \sqrt{\frac{u_0}{10}} \times \Bigg[ &27 \sum_{k=0}^{\infty} \frac{1}{k + \frac{5}{2}} \frac{32}{(21 + 20k + 4k^2)^2} \zeta^{2k+6} \\
&+ (N-1) \sum_{k=0}^{\infty} \frac{1}{k + \frac{3}{2}} \frac{32}{(5 + 12k + 4k^2)^2} \zeta^{2k+4} \Bigg].
\end{aligned}
\tag{94}
$$

Therefore, the prefactors are given by

$$
\begin{aligned}
c_4^L &= \frac{N-1}{400\pi} \sqrt{\frac{u_0}{10}} = \sqrt{\frac{3}{2(3N+22)}} \cdot \frac{N-1}{50} \epsilon^{\frac{1}{2}} + \mathcal{O}(\epsilon), \\
c_k^L &= \frac{3}{16\pi} \frac{N+26}{(k-3)^2(k-1)(k+1)^2} \sqrt{\frac{u_0}{10}} \\
&= \frac{3}{2} \sqrt{\frac{3}{2(3N+22)}} \cdot \frac{N+26}{(k-3)^2(k-1)(k+1)^2} \epsilon^{\frac{1}{2}} + \mathcal{O}(\epsilon), \quad k = 6, 8, 10, \dots
\end{aligned}
$$

We notice that the BOE coefficient $c_4^L$ vanishes when $N = 1$. Analytically, the prefactor $N-1$ arises from the Feynman diagram $c_{24}$ shown in Fig. 3 (b), where the loop is formed by propagators of the transverse modes, yielding the $N-1$ factor. This gives rise to Eq. (83), which includes the higher-order term $H''(\zeta)$. The leading contribution to $H''(\zeta)$ is of order $\zeta^4$. In contrast, the term $H'(\zeta)$ originates from the Feynman diagram $c_{23}$, also shown in Fig. 3 (b), where the loop consists of propagators of the longitudinal modes. Its leading contribution appears at order $\zeta^6$. Hence, $c_4^L$ arises exclusively from the transverse mode, explaining the factor of $N-1$. Once again, we observe that the leading term in the $\epsilon$ expansion is $\mathcal{O}(\epsilon^{1/2})$. We also keep the expression with $u_0$ in the first equality, and in strictly $d = 3$, $u_0$ follows a logarithmic flow, $u_0 \sim \frac{1}{\log \mu_0 z}$.

# 5 Extraordinary phase in tricritical $O(N)$ model

We first briefly review the extraordinary-log transition in the critical $O(N)$ model described in Ref. [11], and then consider the case of the tricritical $O(N)$ model. Readers may also refer to Ref. [37] for a clear and precise argument for the derivation.

## 5.1 IR description of the extraordinary-log transition

Consider a model system, such as a lattice model, featuring the $O(N)$ transition in the bulk. To investigate the extraordinary-log transition in the critical $O(N)$ model, we can imagine

stripping away the outermost boundary layer of the system. The remaining system, without the outermost layer, is described by the ordinary fixed point. On the other hand, because we are interested in the extraordinary transition, the order parameter is ordered in the outermost layer. It can be modeled by a NLSM [38, 39] with $g \sim 0$. We, then, couple the ordinary fixed point action to the NLSM, while respecting the $O(N)$ symmetry. This is the UV description of the extraordinary-log transition [11]:

$$S_{\text{UV}} = S_{\text{ordinary}} + S_n + S_{n\phi} \,, \tag{95}$$

where $S_{\text{ordinary}}$ denotes the bulk action at the ordinary transition. Here, $S_n$ represents the NLSM and $S_{n\phi}$ describes the coupling between the bulk and the outermost layer, given, respectively, by

$$S_n = \int \mathrm{d}^{d-1}r \, \frac{1}{2g}(\partial_\mu \vec{n})^2 \,, \qquad S_{n\phi} = -\tilde{s} \int \mathrm{d}^{d-1}r \, \vec{n}(x) \cdot \vec{\phi}(x) \,, \tag{96}$$

where $\vec{\phi}$ is the bulk $O(N)$ field and $\tilde{s}$ is the coupling between the bulk and boundary fields.[6] Notice that for the NLSM, we have $g \sim 0$ and $\vec{n}^2 = 1$. Without loss of generality, we take $n_1 = 1$ to be the ordered direction, and $n_i = 0$ for $i > 1$.

Because $\vec{n}$ is ordered, it acts as an external field at the boundary that couples to the bulk $O(N)$ field. As a result, we expect that the theory to flow to the normal fixed point. Notably, the normal fixed point is equivalent to the extraordinary fixed point. For later convenience, we describe the bulk-boundary OPE under the normal boundary condition. The bulk-boundary OPE of the critical $O(N)$ field is given by

$$\phi_1(r, z) \sim \frac{a_\sigma}{(2z)^{\Delta_\phi}} + \frac{a'_\sigma}{(2z)^{\Delta_\phi - d}} \hat{D}(r) + \dots \,, \tag{97a}$$

$$\phi_i(r, z) \sim \frac{b_t}{(2z)^{\Delta_\phi - (d-1)}} \hat{t}_i(r) + \dots \,, \quad i > 1 \,. \tag{97b}$$

In the expansion of the longitudinal component $\phi_1$, $a_\sigma$ represents the normalization of the single point function, and $\hat{D}$ is the displacement operator with scaling dimension $d$. In the expansion of the transverse component $\phi_i$, $i > 1$, $\hat{t}_i$ is the "tilt" field, which is the $O(N-1)$ vector with the lowest scaling dimension $d - 1$.

Including the coupling to the NLSM, the IR description of the extraordinary transition is

$$S_{\text{IR}} = S_{\text{normal}} + S_n - s \int \mathrm{d}^{d-1}r \, \sum_{i>1} \pi_i(r) \hat{t}_i(r) + \delta S \,, \tag{98}$$

where $\delta S$ contains irrelevant terms and $\vec{n} = (\sqrt{1 - \vec{\pi}^2}, \vec{\pi})$. $s$ is the IR coupling between the bulk and boundary fields, and crucially, it is fixed by the $O(N)$ symmetry [11].

While the $O(N)$ symmetry is not explicitly manifest in Eq. (98), the theory should respect it. Now consider a small rotation of the ordered direction. For instance, consider a rotation from the first component (which is assumed to be ordered) to the second component by an angle $\alpha \ll 1$. It leads to a nonvanishing one-point function for the second component,

$$\langle \phi_2(0, z) \rangle = \sin \alpha \frac{a_\sigma}{(2z)^{\Delta_\phi}} \approx \alpha \frac{a_\sigma}{(2z)^{\Delta_\phi}} \,. \tag{99}$$

Due to the $O(N)$ symmetry, this result must also be recovered from the IR theory. Because $\vec{n} = (\cos\alpha, \sin\alpha, 0, \dots)$ after the rotation, the second component of the $O(N)$ field, $\phi_2$, has a

---

[6]Note that $\tilde{s}$ is different from $s$ in the IR theory.

nonvanishing expectation value as expected. We calculate this expectation value using the IR theory at the leading order,

$$\langle \phi_2(0,z) \rangle \approx \alpha s \int \mathrm{d}^{d-1} r \langle \phi_2(0,z) \hat{t}_2(r) \rangle_{\text{normal}} \tag{100}$$

$$= \alpha s \int \mathrm{d}^{d-1} r \frac{b_t (2z)^{d-1-\Delta_\phi}}{(r^2 + z^2)^{d-1}} = \alpha s b_t \frac{(4\pi)^{\frac{d-1}{2}} \Gamma(\frac{d-1}{2})}{\Gamma(d-1)} \frac{1}{(2z)^{\Delta_\phi}} . \tag{101}$$

Equating this to Eq. (99), we determine $s$ [11],

$$s = \frac{\Gamma(d-1)}{(4\pi)^{\frac{d-1}{2}} \Gamma(\frac{d-1}{2})} \frac{a_\sigma}{b_t} . \tag{102}$$

Hence, the IR coupling $s$ is fixed by the $O(N)$ symmetry, because $a_\sigma$ and $b_t$ on the right-hand side are universal constants at the fixed point.

After fixing the IR coupling between the bulk and boundary fields, we now examine the effect of such a coupling on the RG equation of the NLSM. The lowest order RG can be obtained straightforwardly [11],

$$\frac{\mathrm{d}g}{\mathrm{d}\log\mu} = \beta_g , \qquad \beta_g = -\frac{N-2}{2\pi} g^2 + \frac{\pi s^2}{2} g^2 , \tag{103}$$

where the first term in the beta equation $\frac{N-2}{2\pi} g^2$ is a well-known result from the NLSM, while the second term arises from the coupling to the bulk field.

In the conventional NLSM without the bulk degrees of freedom, for $N > 2$, the system flows to a strongly coupled fixed point at IR that restores the symmetry, presenting a short-range correlated state. For $N = 2$, the one-loop RG equation vanishes, and the system exhibits a quasi-long range order. This is the seminal result of the Mermin-Wagner theorem. However, the presence of the second term can reverse the RG flow to a weakly coupled fixed point, $g \sim 0$, when $N < N_c$, with

$$N_c = 2 + \pi^2 s^2 . \tag{104}$$

The logarithmic flow to this weakly coupled fixed point leads to a logarithmic correlation of the boundary field. One should notice that the one-point function also decays logarithmically, indicating that no true long-range order has developed. Because $a_\sigma, b_t > 0$, the above analysis implies $N_c > 2$. A conformal bootstrap study of the normal fixed point showed that $N_c > 3$ [37] for the critical $O(N)$ model.

## 5.2 Extraordinary transition at the tricritical point

The above analysis reveals that the RG invariant quantity $s$ crucially determines the critical number of components. As $s$ is fixed by the bulk-boundary OPE of the $O(N)$ field, it can be determined through the BOE of layer susceptibility. First, we determine the value of $s$ in $d = 3 - \epsilon$ calculation. recall that the order parameter profile $m(z)$ corresponds to the one-point function of the longitudinal component. Hence, at the one-loop level, $a_\sigma$ is given by Eq. (62). Next, we extract $b_t$ from the BOE of the two-point function. According to the BOE of layer susceptibility listed in Sec. 4.1, the BOE of the two-point function reads,

$$G^T(r, z, z') = (4zz')^{-\Delta_\phi} c_{d-1} \sigma_{d-1} \mathcal{G}_{\text{boe}}(d-1, \xi) + \dots$$
$$= (4zz')^{-\Delta_\phi} c_{d-1}^T \sigma_{d-1} \xi^{d-1} + \dots , \qquad \xi \to \infty . \tag{105}$$

Comparing this with the bulk-boundary OPE in Eq. (97b), we obtain

$$b_t^2 = c_{d-1}^T \sigma_{d-1} = \frac{1}{16\pi} - \frac{2 + 3c_{1,1} - 3c_{2,1}}{64\pi} \sqrt{\frac{3}{2(3N+22)}} \epsilon^{\frac{1}{2}} + \mathcal{O}(\epsilon). \qquad (106)$$

Hence, we compute $s$ from Eq. (102) for the tricritical point:

$$s^2 = \frac{1}{2\pi^2} \sqrt{\frac{3(3N+22)}{2}} \epsilon^{-\frac{1}{2}} + \frac{78 + 8N + 9c_{1,1} - 9c_{2,1}}{16\pi^2} + \mathcal{O}(\epsilon^{\frac{1}{2}}). \qquad (107)$$

Naively, this suggests that the critical number $N_c$ for the tricritical $O(N)$ model in three dimensions is infinite as $\epsilon \to 0$. However, it corresponds to a $3 - \epsilon$ dimensional bulk theory couples to a two-dimensional boundary, which does not represent the standard codimensional-one boundary.[7] Therefore, to properly handle the BCFT at $d = 3$, we shall incorporate the RG equations for both $s$ and $g$ on an equal footing for $\epsilon = 0$. To this end, considering the coupling $s$ given in Eq. (102), and the logarithmic flow of BOE coefficient $a_\sigma$ discussed below Eq. (62), we can get the following coupled RG equations:

$$\frac{ds}{d\log\mu} = -\frac{3(3N+22)}{128\pi^4} s^{-3}, \qquad (108)$$

$$\frac{dg}{d\log\mu} = -\frac{N-2}{2\pi} g^2 + \frac{\pi s^2}{2} g^2. \qquad (109)$$

The solution in the long-wave length limit is[8]

$$s \approx \left[ \frac{3(3N+22)}{2(2\pi)^4} \log\mu_0 l \right]^{1/4}, \quad g \approx \frac{48(N-2)\pi}{3N+22} (\log\mu_0 l)^{-2} + \sqrt{\frac{96\pi^2}{3N+22}} (\log\mu_0 l)^{-3/2}, \qquad (110)$$

where $\mu_0$ is an arbitrary energy scale, and $l$ denotes the length scale of the system. The IR limit corresponds to the long wavelength regime, $l \to \infty$. It is important to note that the product $sg$ flows to zero in this limit, thereby justifying the use of perturbative methods. Namely, the RG calculation in Eq. (103) relies on a perturbative expansion in the parameter $sg$, which is obvious in the second term of the beta function of the coupling $g$.

Now we can solve the RG equation for the renormalized correlation function,

$$D_r^m = Z_n^{-m/2} \langle n(x_1) \dots n(x_m) \rangle, \qquad (111)$$

where $Z_n$ is the RG factor for the field $n$, and satisfies the equation [11]

$$\mu \frac{\partial}{\partial \mu} \log Z_n = \eta(g), \qquad \eta(g) = \frac{N-1}{2\pi} g. \qquad (112)$$

The RG equation is given by

$$\left( \mu \frac{\partial}{\partial \mu} + \beta(g) \frac{\partial}{\partial g} + \frac{m}{2} \eta(g) \right) D_r^m(g, \mu) = 0. \qquad (113)$$

Here, the beta equation $\beta_g$ involves a logarithmic flow of the bulk-boundary coupling $s$, in contrast to the critical $O(N)$ model, where the bulk-boundary coupling is a constant. Thus, incorporating the logarithmic flow of $s$, the modified RG equation is

$$\left( \mu \frac{\partial}{\partial \mu} + \left( \frac{\pi}{2} \sqrt{\frac{3(3N+22)}{2(2\pi)^4}} \log\mu - \frac{N-2}{2\pi} \right) g^2 \frac{\partial}{\partial g} + \frac{m}{2} \frac{N-1}{2\pi} g \right) D_r^m = 0. \qquad (114)$$

---

[7]We thank the referee for pointing this out.

[8]The full solution is provided in Appendix C, where we show that, for the fixed point discussed below, the perturbative calculation is self-consistent.

From the RG equation, we can get the expectation value of the surface order,

$$\langle n \rangle_r \sim \exp\left[ (N-1)\sqrt{\frac{24}{3N+22}} \left( \log \frac{\mu_0}{\sqrt{h}} \right)^{-1/2} \right], \qquad h \to 0, \tag{115}$$

where $h$ represents an external field linearly coupled to the surface order $\vec{n}$, while $\mu_0$ is an arbitrary energy scale. We observe that as the external field $h$ approaches zero, the surface order parameter remains finite and does not vanish. Also, the correlation function is dominated by the finite orders in this case,

$$\langle n(x)n(0) \rangle_r \sim \exp\left[ 4(N-1)\sqrt{\frac{6}{3N+22}} (\log \mu_0 x)^{-1/2} \right], \qquad x \gg 1. \tag{116}$$

On top of the ordered configuration, the correlations are expected to be governed by the leading boundary primary fields, namely, the displacement field in the longitudinal mode and the tilt field in the transverse mode.

It is illuminating to provide a general discussion of the logarithmic flow of the coupling constant $g$ and the boundary order. Although the coupling constant $g$ flows to zero logarithmically in both the extraordinary-log transition of the critical $O(N)$ model and the extraordinary transition of the tricritical $O(N)$ model discussed above, whether the boundary order parameter vanishes depends on the specific form (or order) of the logarithmic flow. Consider a general logarithmic flow in the long-wavelength limit $\mu \ll \mu_0$ of the form

$$g(\mu) \approx \left( \log \frac{\mu_0}{\mu} \right)^{-m}, \tag{117}$$

where we neglect any constant prefactor for simplicity. The coupling exhibits a logarithmic flow to zero with an exponent $m > 0$. Then, the RG factor, defined via $n = Z_n^{-1/2} n_0$, determines the flow of the surface order,

$$\mu \frac{\mathrm{d}}{\mathrm{d}\mu} n(\mu) = \frac{N-1}{2\pi} g(\mu) n(\mu), \tag{118}$$

where we use the anomalous dimension in Eq. (112). This yields the following solution for the surface order parameter

$$n(\mu) \approx \begin{cases} \exp\left[ -\frac{N-1}{2\pi} \frac{1}{1-m} \left( \log \frac{\mu_0}{\mu} \right)^{1-m} \right], & m \neq 1, \\ \left( \frac{N-1}{2\pi} \log \frac{\mu_0}{\mu} \right)^{-1}, & m = 1. \end{cases} \tag{119}$$

If $g$ decays slowly with $m \leq 1$, the surface order parameter vanishes in the low-energy limit. In particular, it exhibits a logarithmic flow to zero when $m = 1$, which corresponds exactly to the case of extraordinary-log criticality. Only when $g$ decays fast enough, with $m > 1$, does the surface order remain finite in the long-wavelength limit, i.e., the surface remains ordered. This occurs at the extraordinary transition in the tricritical model, since $m > 1$ in the RG flow given in Eq. (110).

Lastly, we comment on the relation between our result and the no-go theorem for continuous symmetry breaking in the context of BCFT by Cuomo and Zhang in Ref. [25]. Cuomo and Zhang analyzed the Ward identity for a continuous internal symmetry (the $O(N)$ symmetry in our case) in BCFT. The Ward identity can be decomposed into a bulk part and a boundary part. With a crucial assumption that the RG terminates at a fixed point, continuous symmetry breaking would lead to a pole in the boundary spectral weight, which implies a decoupled sector

of gapless Goldstone modes. Then, the machinery of M-W-C theorem can be applied to argue the absence of symmetry breaking on the surface. In our case, this assumption does not hold. More specifically, the marginal logarithmic flow for the bulk coupling constant couples to the boundary flow of the NLSM, which then renders an intriguing logarithmic flow with $m > 1$ for the NLSM coupling $g$. If we assume that the bulk RG terminates at the fixed point, specifically a Gaussian fixed point, then the mean-field order parameter profile Eq. (49) and the mean-field Green's function Eq. (51) would no longer be valid. This would imply that the extraordinary transition does not exist, a conclusion that is widely regarded as incorrect [26, 27]. Hence, it is crucial to treat the boundary and bulk RG on equal footing to reach a correct conclusion, thereby falsifying the assumption in Ref. [25].

# 6 Conclusion

In this work, we extend the understanding of the extraordinary transition in three-dimensional $O(N)$ boundary conformal field theories, focusing on the tricritical $O(N)$ model. By deriving the mean-field order parameter profile and propagator for general $|\vec{\phi}|^{2n}$ couplings and employing the technique of layer susceptibility, we construct the boundary operator expansion (BOE) for the tricritical case ($n = 3$) at the one-loop level. Our results reveal the boundary operator spectrum and explore the extraordinary transition beyond the $O(N)$ model, demonstrating that an ordered boundary exists for any $N$ in three dimensions. This exemplifies a scenario of continuous symmetry breaking in two dimensions under boundary criticality, in stark contrast to the critical $O(N)$ model with vanishing boundary orders. It arises because the upper critical dimension for the tricritical model is $d = 3$, leading to milder fluctuations compared to the critical $O(N)$ case.

Tricritical points emerge in a wide range of physical systems, including ferroelectrics [40], liquid crystal [41–43], and polymer solutions [17, 44]. Interestingly, the $\theta$-point of a long polymer chain corresponds to the tricritical point of a vector field theory with zero components ($N = 0$) [45–47]. When these systems are confined by a boundary, which is unavoidable in the real world, it leads to fruitful boundary critical behaviors. While the ordinary and special transitions have been explored in Ref. [19], our work addresses the gaps in understanding the extraordinary transition. Realizations of the bulk tricritical point have been reported in Ref. [48,49], also numerically verified in Ref. [50–52]. It is worth emphasizing that the extraordinary transition requires no additional fine-tuning beyond the bulk tricritical point. Hence, we expect that our results can be experimentally tested. Finally, the methods developed in this study lay a foundation for exploring BOEs in other models, and inspire investigations into boundary criticality at the tricritical point. More broadly, we believe that this work will motivate further investigations into boundary criticality and its implications across diverse physical systems, such as topological systems [53–63] and holographic systems [64–71]. For example, the nature of the extraordinary transition in chiral tricritical point in topological systems [24] remains an open question. Moreover, while this work focuses on the weakly coupled regime, it would be interesting to explore strongly interacting BCFTs with nontrivial extraordinary behavior using holographic techniques.

# Acknowledgments

**Funding information** X. S. acknowledges the support from the Lavin-Bernick Grant during his visit to Tulane University, where the work was conducted. The work of S.-K. J. is supported by a start-up grant and a COR Research Fellowship from Tulane University.

# A   Properties of special functions

We summarize the properties of Bessel and Hypergeometric functions used in the evaluation of the Feynman diagrams.

## A.1   Determination of integral constants $C_1^{L,T}$

Defining $Z_\nu(x) = e^{i\pi\nu}K_\nu(x)$, where $K_\nu$ is the Bessel equation of the second kind, it has the following properties [72]

$$
\begin{aligned}
Z_{\nu-1}(x) - Z_{\nu+1}(x) &= \frac{2\nu}{x}Z_\nu(x), \\
Z_{\nu-1}(x) + Z_{\nu+1}(x) &= 2Z_\nu'(x) = \frac{2\nu}{x}Z_\nu(x) + 2Z_{\nu+1}(x).
\end{aligned}
\tag{A.1}
$$

Therefore, the second equation gives $e^{-i\pi}K_{\nu-1}(x) + e^{i\pi}K_{\nu+1}(x) = \frac{2\nu}{x}K_\nu(x) + 2e^{i\pi}K_{\nu+1}(x)$. Then, Eq. (42) can be simplified to

$$
1 = -C_L p e^{-i\frac{n\pi}{1-n}} i x \left[ K_{\alpha_L}(x)K_{\alpha_L+1}(-x) + K_{\alpha_L}(-x)K_{\alpha_L+1}(x) \right].
\tag{A.2}
$$

Furthermore, defining $K_\nu(x) = \frac{1}{2}\pi i e^{i\frac{\nu\pi}{2}} H_\nu^{(1)}(e^{i\frac{\pi}{2}}x)$ and $K_\nu(e^{i\pi}x) = -\frac{1}{2}\pi i e^{-i\frac{\nu\pi}{2}} H_\nu^{(2)}(e^{i\frac{\pi}{2}}x)$, where $H_\nu^{(1,2)}(z)$ are Henkel functions, we have the property for $H_\nu^{(1)}$ and $H_\nu^{(2)}$ that [72]

$$
H_{\nu+1}^{(1)}(x)H_\nu^{(2)}(x) - H_\nu^{(1)}(x)H_{\nu+1}^{(2)}(x) = -\frac{4i}{\pi x}.
\tag{A.3}
$$

With this, Eq. (42) can be simplified to

$$
\begin{aligned}
1 &= -C_L p e^{-i\frac{n\pi}{1-n}} i x \frac{\pi^2}{4} \left[ e^{-i\frac{\pi}{2}} H_{\alpha_L}^{(1)}(e^{i\frac{\pi}{2}}x)H_{\alpha_L+1}^{(2)}(e^{i\frac{\pi}{2}}x) + e^{i\frac{\pi}{2}} H_{\alpha_L}^{(2)}(e^{i\frac{\pi}{2}}x)H_{\alpha_L+1}^{(1)}(e^{i\frac{\pi}{2}}x) \right] \\
&= C_L p e^{-i\frac{n\pi}{1-n}} x \frac{\pi^2}{4} \frac{-4i}{\pi x e^{i\frac{\pi}{2}}} = -\pi C_L p e^{-i\frac{n\pi}{1-n}}.
\end{aligned}
\tag{A.4}
$$

## A.2   Properties of regularized generalized hypergeometric function

The regularized generalized hypergeometric function $_p\tilde{F}_q[a,b,z]$ is related to the generalized hypergeometric function $_pF_q[a,b,z]$ normalization with the gamma functions:

$$
_p\tilde{F}_q[a,b,z] = \frac{_pF_q[a,b,z]}{\Gamma(b_1)\cdots\Gamma(b_q)}, \qquad a = \{a_1,\cdots,a_p\}, \qquad b = \{b_1,\cdots,b_q\}.
\tag{A.5}
$$

More explicitly, the regularized generalized hypergeometric function is defined as

$$
_p\tilde{F}_q[a,b,z] = \sum_{k=0}^{\infty} \frac{\prod_{j=1}^{p}(a_j)_k}{\Gamma(k+1)\prod_{i=1}^{q}\Gamma(b_i+k)} z^k,
\tag{A.6}
$$

where $(a_j)_k = a_j(a_j+1)\ldots(a_j+k-1) = \frac{\Gamma(a_j+k)}{\Gamma(a_j)}$ is the Pochhammer symbols.

In the main text, we make use of the following integral identity involving the regularized generalized hypergeometric function:

$$
\int dz\, z^{\alpha-1}\, _p\tilde{F}_q\left[\{a_1,\ldots,a_p\},\{b_1,\ldots,b_q\},z\right] = \Gamma(\alpha)z^\alpha\, _{p+1}\tilde{F}_{q+1}\left[\{\alpha,a_1,\ldots,a_p\},\{\alpha+1,b_1,\ldots,b_q\},z\right].
\tag{A.7}
$$

# B  Details of Feynman diagram calculations

In this section, we show the details for the calculation of the one-point and two-point functions in the main text.

## B.1  Details of the evaluation for the integral $\int \mathrm{d}p\, p^{d-2} I_\nu(pz) I_{\nu+1}(pz) K_\nu(pz') K_{\nu+1}(pz')$

First, we expand $I_\nu(pz) I_{\nu+1}(pz)$ as a power series in $pz$ [72]

$$I_\nu(pz) I_{\nu+1}(pz) = \left(\frac{1}{2}pz\right)^{2\nu+1} \sum_{k=0}^{\infty} \frac{(2\nu+k+2)_k (\frac{1}{2}pz)^{2k}}{k!\,\Gamma(\nu+k+1)\Gamma(\nu+k+2)}, \tag{B.1}$$

where $(a)_k = \Gamma(a+k)/\Gamma(a)$. Next, we perform the integration term by term and then sum over $k$, which yields

$$\int \mathrm{d}p\, p^{d-2} I_\nu(pz) I_{\nu+1}(pz) K_\nu(pz') K_{\nu+1}(pz') \tag{B.2}$$

$$= \sum_{k=0}^{\infty} \frac{(2\nu+k+2)_k (\frac{1}{2}z)^{2k+2\nu+1}}{k!\,\Gamma(\nu+k+1)\Gamma(\nu+k+2)} \int \mathrm{d}p\, p^{d-2+2k+2\nu+1} K_\nu(pz') K_{\nu+1}(pz')$$

$$= \sum_{k=0}^{\infty} \frac{(2\nu+k+2)_k (\frac{1}{2}z)^{2k+2\nu+1}}{k!\,\Gamma(\nu+k+1)\Gamma(\nu+k+2)} z'^{-d-2k-2\nu}$$

$$\times \frac{\sqrt{\pi}\,\Gamma(\frac{d-2+2k+2\nu+1}{2} - \nu)\Gamma(\frac{d-2+2k+2\nu+1}{2} + \nu + 1)\Gamma(\frac{d-2+2k+2\nu+1}{2})}{4\Gamma(\frac{1+d-2+2k+2\nu+1}{2})}$$

$$= z'^{1-d} \frac{1}{4} \left(\frac{z}{z'}\right)^{1+2\nu} \Gamma\left(\frac{d-1}{2}\right)\Gamma\left(\frac{3+2\nu}{2}\right)\Gamma\left(\frac{d-1+2\nu}{2}\right)\Gamma\left(\frac{d+1+4\nu}{2}\right)$$

$$\times\, {}_4\tilde{F}_3\left[\left\{\frac{d-1}{2}, \frac{3+2\nu}{2}, \frac{d-1+2\nu}{2}, \frac{d+1+4\nu}{2}\right\}, \left\{2+\nu, \frac{d+2\nu}{2}, 2+2\nu\right\}, \left(\frac{z}{z'}\right)^2\right],$$

where ${}_p\tilde{F}_q[a, b, z]$ is the regularized generalized hypergeometric function, ${}_p\tilde{F}_q[a, b, z] = \frac{{}_pF_q[a,b,z]}{\Gamma(b_1)\cdots\Gamma(b_q)}$, with $a = \{a_1, \cdots, a_p\}$ and $b = \{b_1, \cdots, b_q\}$.

## B.2  Details of the evaluation for the integral $I_1$ and $I_2$

We present the detail of the evaluation for the integral $I_1$ and $I_2$ in Eq. (70). The definition of the integral $I_{1,2}$ is given in the first equality in the following equations. We subsequently make a coordinate transformation $Y = \frac{y_2}{y_1}$ in $I_1$ and $Y = \frac{y_1}{y_2}$ in $I_2$:

$$I_1 = \int_0^\infty \mathrm{d}y_1 \int_{y_1}^\infty \mathrm{d}y_2 \frac{1}{y_1 y_2} \frac{\min(z_1, y_1)}{\max(z_1, y_1)} \frac{\min(z_2, y_2)}{\max(z_2, y_2)} y_1^{3-d} \left(\frac{y_1}{y_2}\right)^{d+1}$$

$$\times\, {}_4\tilde{F}_3\left[\left\{\frac{d-1}{2}, \frac{5}{2}, \frac{d+1}{2}, \frac{d+5}{2}\right\}, \left\{3, \frac{d+2}{2}, 4\right\}, \left(\frac{y_1}{y_2}\right)^2\right]$$

$$= \int_1^\infty \mathrm{d}Y\, Y^{-d-2}\, {}_4\tilde{F}_3\left[\left\{\frac{d-1}{2}, \frac{5}{2}, \frac{d+1}{2}, \frac{d+5}{2}\right\}, \left\{3, \frac{d+2}{2}, 4\right\}, Y^{-2}\right]$$

$$\times \int_0^\infty \mathrm{d}y_1 y_1^{2-d} \frac{\min(z_1, y_1)}{\max(z_1, y_1)} \frac{\min(\frac{z_2}{Y}, y_1)}{\max(\frac{z_2}{Y}, y_1)},$$

$$I_2 = \int_0^\infty dy_2 \int_{y_2}^\infty dy_1 \frac{1}{y_1 y_2} \frac{\min(z_1, y_1)}{\max(z_1, y_1)} \frac{\min(z_2, y_2)}{\max(z_2, y_2)} y_2^{3-d} \left(\frac{y_2}{y_1}\right)^{d+1}$$

$$\times {}_4\tilde{F}_3 \left[ \left\{ \frac{d-1}{2}, \frac{5}{2}, \frac{d+1}{2}, \frac{d+5}{2} \right\}, \left\{ 3, \frac{d+2}{2}, 4 \right\}, (\frac{y_2}{y_1})^2 \right]$$

$$= \int_1^\infty dY \, Y^{-d-2} \, {}_4\tilde{F}_3 \left[ \left\{ \frac{d-1}{2}, \frac{5}{2}, \frac{d+1}{2}, \frac{d+5}{2} \right\}, \left\{ 3, \frac{d+2}{2}, 4 \right\}, Y^{-2} \right]$$

$$\times \int_0^\infty dy_2 \, y_2^{2-d} \frac{\min(\frac{z_1}{Y}, y_2)}{\max(\frac{z_1}{Y}, y_2)} \frac{\min(z_2, y_2)}{\max(z_2, y_2)}.$$

In the following, we give the derivation for $z_1 < z_2$. The result for $z_1 > z_2$ can be obtained by exchanging $z_1$ and $z_2$.

For $z_1 < z_2$, we can directly perform the following integration,

$$\int_0^\infty dy \, y^{2-d} \frac{\min(z_1, y)}{\max(z_1, y)} \frac{\min(z_2, y)}{\max(z_2, y)} = z_1^{3-d} (A\zeta + B\zeta^{d-2}), \tag{B.3}$$

with $A = \frac{-2}{(d-5)(d-3)}$, $B = \frac{-2}{(d-3)(d-1)}$. Note that $\zeta = \frac{\min(z_1, z_2)}{\max(z_1, z_2)} = \frac{z_1}{z_2} < 1$ in the above expression as $z_1 < z_2$, so the integral $I_1$ can be simplified as

$$I_1 = \int_1^\infty dY \, Y^{-d-2} \, {}_4\tilde{F}_3 \left[ \left\{ \frac{d-1}{2}, \frac{5}{2}, \frac{d+1}{2}, \frac{d+5}{2} \right\}, \left\{ 3, \frac{d+2}{2}, 4 \right\}, Y^{-2} \right] \tag{B.4}$$

$$\times \left( \Theta(\zeta^{-1} - Y) z_1^{3-d} [A(\zeta Y) + B(\zeta Y)^{d-2}] + \Theta(Y - \zeta^{-1}) \left(\frac{z_2}{Y}\right)^{3-d} [A(\zeta Y)^{-1} + B(\zeta Y)^{-d+2}] \right)$$

$$= \frac{z_1^{3-d}}{2} \int_{\zeta^2}^1 dy \, y^{\frac{d-1}{2}} \, {}_4\tilde{F}_3 \left[ \left\{ \frac{d-1}{2}, \frac{5}{2}, \frac{d+1}{2}, \frac{d+5}{2} \right\}, \left\{ 3, \frac{d+2}{2}, 4 \right\}, y \right] \left( A\zeta y^{-\frac{1}{2}} + B\zeta^{d-2} y^{-\frac{d-2}{2}} \right)$$

$$+ \frac{z_2^{3-d}}{2} \int_0^{\zeta^2} dy \, y^{\frac{d-1}{2} + \frac{3-d}{2}} \, {}_4\tilde{F}_3 \left[ \left\{ \frac{d-1}{2}, \frac{5}{2}, \frac{d+1}{2}, \frac{d+5}{2} \right\}, \left\{ 3, \frac{d+2}{2}, 4 \right\}, y \right]$$

$$\times \left( A\zeta^{-1} y^{\frac{1}{2}} + B\zeta^{-d+2} y^{\frac{d-2}{2}} \right),$$

where we have made a variable transformation $y = Y^{-2}$ in the second equality. To calculate it explicitly, we use the following property of regularized generalized hypergeometric function [73]

$$\int dz \, z^{\alpha-1} \, {}_p\tilde{F}_q[\{a_1, \cdots, a_p\}, \{b_1, \cdots, b_q\}, z] = \Gamma(\alpha) z^\alpha \, {}_{p+1}\tilde{F}_{q+1}[\{\alpha, a_1, \cdots, a_p\}, \{\alpha+1, b_1, \cdots, b_q\}, z]. \tag{B.5}$$

Therefore, the integral $I_1$ becomes

$$I_1 = \frac{z_1^{3-d}}{2} A\zeta \left\{ \Gamma\left(\frac{d}{2}\right) y^{\frac{d}{2}} \, {}_5\tilde{F}_4 \left[ \left\{ \frac{d}{2}, \frac{d-1}{2}, \frac{5}{2}, \frac{d+1}{2}, \frac{d+5}{2} \right\}, \left\{ \frac{d+2}{2}, 3, \frac{d+2}{2}, 4 \right\}, y \right] \right\}_{\zeta^2}^1 \tag{B.6}$$

$$+ \frac{z_1^{3-d}}{2} B\zeta^{d-2} \left\{ \Gamma\left(\frac{3}{2}\right) y^{\frac{3}{2}} \, {}_5\tilde{F}_4 \left[ \left\{ \frac{3}{2}, \frac{d-1}{2}, \frac{5}{2}, \frac{d+1}{2}, \frac{d+5}{2} \right\}, \left\{ \frac{5}{2}, 3, \frac{d+2}{2}, 4 \right\}, y \right] \right\}_{\zeta^2}^1$$

$$+ \frac{z_2^{3-d}}{2} A\zeta^{-1} \left\{ \Gamma\left(\frac{5}{2}\right) y^{\frac{5}{2}} \, {}_5\tilde{F}_4 \left[ \left\{ \frac{5}{2}, \frac{d-1}{2}, \frac{5}{2}, \frac{d+1}{2}, \frac{d+5}{2} \right\}, \left\{ \frac{7}{2}, 3, \frac{d+2}{2}, 4 \right\}, y \right] \right\}_0^{\zeta^2}$$

$$+ \frac{z_2^{3-d}}{2} B\zeta^{-d+2}$$

$$\times \left\{ \Gamma\left(\frac{d+2}{2}\right) y^{\frac{d+2}{2}} \, {}_5\tilde{F}_4 \left[ \left\{ \frac{d+2}{2}, \frac{d-1}{2}, \frac{5}{2}, \frac{d+1}{2}, \frac{d+5}{2} \right\}, \left\{ \frac{d+4}{2}, 3, \frac{d+2}{2}, 4 \right\}, y \right] \right\}_0^{\zeta^2},$$

where $\{f(y)\}_a^b = f(b) - f(a)$. Using the same method, the integral $I_2$ in Eq. (B.3) can be evaluated. For brevity, we omit the explicit derivation. Now, after both integrals have been evaluated, the final result can be simplified further to get

$$
\begin{aligned}
I_1 + I_2 = \frac{z_1^{3-d}}{2} A\zeta &\left\{ \Gamma\left(\frac{d}{2}\right) {}_5\tilde{F}_4\left[\left\{\frac{d}{2}, \frac{d-1}{2}, \frac{5}{2}, \frac{d+1}{2}, \frac{d+5}{2}\right\}, \left\{\frac{d+2}{2}, 3, \frac{d+2}{2}, 4\right\}, 1\right]\right. \\
&\left. + \Gamma\left(\frac{5}{2}\right) {}_5\tilde{F}_4\left[\left\{\frac{5}{2}, \frac{d-1}{2}, \frac{5}{2}, \frac{d+1}{2}, \frac{d+5}{2}\right\}, \left\{\frac{7}{2}, 3, \frac{d+2}{2}, 4\right\}, 1\right]\right\} \\
+ \frac{z_1^{3-d}}{2} B\zeta^{d-2} &\left\{ \Gamma\left(\frac{3}{2}\right) {}_5\tilde{F}_4\left[\left\{\frac{3}{2}, \frac{d-1}{2}, \frac{5}{2}, \frac{d+1}{2}, \frac{d+5}{2}\right\}, \left\{\frac{5}{2}, 3, \frac{d+2}{2}, 4\right\}, 1\right]\right. \\
&\left. + \Gamma\left(\frac{d+2}{2}\right) {}_5\tilde{F}_4\left[\left\{\frac{d+2}{2}, \frac{d-1}{2}, \frac{5}{2}, \frac{d+1}{2}, \frac{d+5}{2}\right\}, \left\{\frac{d+4}{2}, 3, \frac{d+2}{2}, 4\right\}, 1\right]\right\} \\
+ \frac{z_1^{3-d}}{2} A\zeta^{d+1} &\left\{ \Gamma\left(\frac{5}{2}\right) {}_5\tilde{F}_4\left[\left\{\frac{5}{2}, \frac{d-1}{2}, \frac{5}{2}, \frac{d+1}{2}, \frac{d+5}{2}\right\}, \left\{\frac{7}{2}, 3, \frac{d+2}{2}, 4\right\}, \zeta^2\right]\right. \\
&\left. - \Gamma\left(\frac{d}{2}\right) {}_5\tilde{F}_4\left[\left\{\frac{d}{2}, \frac{d-1}{2}, \frac{5}{2}, \frac{d+1}{2}, \frac{d+5}{2}\right\}, \left\{\frac{d+2}{2}, 3, \frac{d+2}{2}, 4\right\}, \zeta^2\right]\right\} \\
+ \frac{z_1^{3-d}}{2} B\zeta^{d+1} &\left\{ \Gamma\left(\frac{d+2}{2}\right) {}_5\tilde{F}_4\left[\left\{\frac{d+2}{2}, \frac{d-1}{2}, \frac{5}{2}, \frac{d+1}{2}, \frac{d+5}{2}\right\}, \left\{\frac{d+4}{2}, 3, \frac{d+2}{2}, 4\right\}, \zeta^2\right]\right. \\
&\left. - \Gamma\left(\frac{3}{2}\right) {}_5\tilde{F}_4\left[\left\{\frac{3}{2}, \frac{d-1}{2}, \frac{5}{2}, \frac{d+1}{2}, \frac{d+5}{2}\right\}, \left\{\frac{5}{2}, 3, \frac{d+2}{2}, 4\right\}, \zeta^2\right]\right\}.
\end{aligned}
\tag{B.7}
$$

The result for $z_1 > z_2$ can be obtained by exchanging $z_1$ and $z_2$. Incorporating this result and organizing the expression in powers of $\zeta$, Eq. (70) simplifies to:

$$
b'_{23} = \frac{90^{\frac{3}{2}}}{(6!)^2} \frac{S_{d-1}}{32(2\pi)^{d-1}} \sqrt{u_0}(z_1 z_2)^{\frac{4-d}{2}} \left(AC_1 \zeta^{\frac{5-d}{2}} + BC_2 \zeta^{\frac{d-1}{2}}\right) + (z_1 z_2)^{\frac{4-d}{2}} H(\zeta),
\tag{B.8}
$$

where the constants $C_1$ and $C_2$ are

$$
\begin{aligned}
C_1 = &\Gamma\left(\frac{d-1}{2}\right)\Gamma\left(\frac{5}{2}\right)\Gamma\left(\frac{d+1}{2}\right)\Gamma\left(\frac{d+5}{2}\right) \\
&\times \left\{\Gamma\left(\frac{d}{2}\right) {}_5\tilde{F}_4\left[\left\{\frac{d}{2}, \frac{d-1}{2}, \frac{5}{2}, \frac{d+1}{2}, \frac{d+5}{2}\right\}, \left\{\frac{d+2}{2}, 3, \frac{d+2}{2}, 4\right\}, 1\right]\right. \\
&\left. + \Gamma\left(\frac{5}{2}\right) {}_5\tilde{F}_4\left[\left\{\frac{5}{2}, \frac{d-1}{2}, \frac{5}{2}, \frac{d+1}{2}, \frac{d+5}{2}\right\}, \left\{\frac{7}{2}, 3, \frac{d+2}{2}, 4\right\}, 1\right]\right\}, \\
C_2 = &\Gamma\left(\frac{d-1}{2}\right)\Gamma\left(\frac{5}{2}\right)\Gamma\left(\frac{d+1}{2}\right)\Gamma\left(\frac{d+5}{2}\right) \\
&\times \left\{\Gamma\left(\frac{3}{2}\right) {}_5\tilde{F}_4\left[\left\{\frac{3}{2}, \frac{d-1}{2}, \frac{5}{2}, \frac{d+1}{2}, \frac{d+5}{2}\right\}, \left\{\frac{5}{2}, 3, \frac{d+2}{2}, 4\right\}, 1\right]\right. \\
&\left. + \Gamma\left(\frac{d+2}{2}\right) {}_5\tilde{F}_4\left[\left\{\frac{d+2}{2}, \frac{d-1}{2}, \frac{5}{2}, \frac{d+1}{2}, \frac{d+5}{2}\right\}, \left\{\frac{d+4}{2}, 3, \frac{d+2}{2}, 4\right\}, 1\right]\right\},
\end{aligned}
\tag{B.9}
$$

and the last term is given in Eq. (73) in the main text.

## B.3  Details of the evaluation for the integral $I'_{1,2}$ and $I''_{1,2}$

The evaluation of the integrals $I'_{1,2}$ follows the same procedure as for $I_{1,2}$. Through a series of variable changes and simplifications, the integral can be reduced to

$$
\begin{aligned}
I'_1 = \int_0^\infty dy_1 \int_{y_1}^\infty dy_2 \frac{1}{y_1 y_2} &\left(\frac{\min(z_1, y_1)}{\max(z_1, y_1)}\right)^2 \left(\frac{\min(z_2, y_2)}{\max(z_2, y_2)}\right)^2 y_1^{3-d}\left(\frac{y_1}{y_2}\right)^{d+2} \\
&\times {}_4\tilde{F}_3\left[\left\{\frac{5}{2}, \frac{d-1}{2}, \frac{d+3}{2}, \frac{d+7}{2}\right\}, \left\{3, 5, \frac{d+4}{2}\right\}, \left(\frac{y_1}{y_2}\right)^2\right]
\end{aligned}
\tag{B.10}
$$

$$= \frac{z_1^{3-d}}{2} \int_{\zeta^2}^1 dy\, y^{\frac{d}{2}}\, {}_4\tilde{F}_3\left[\left\{\frac{5}{2}, \frac{d-1}{2}, \frac{d+3}{2}, \frac{d+7}{2}\right\}, \left\{3, 5, \frac{d+4}{2}\right\}, y\right]\left(A'\zeta^{d-1}y^{-\frac{d-1}{2}} + B'\zeta^2 y^{-1}\right)$$

$$+ \frac{z_2^{3-d}}{2} \int_0^{\zeta^2} dy\, y^{\frac{3}{2}}\, {}_4\tilde{F}_3\left[\left\{\frac{5}{2}, \frac{d-1}{2}, \frac{d+3}{2}, \frac{d+7}{2}\right\}, \left\{3, 5, \frac{d+4}{2}\right\}, y\right]\left(A'\zeta^{1-d}y^{\frac{d-1}{2}} + B'\zeta^{-2}y\right),$$

where, in the derivation, we have assumed $z_1 < z_2$. Using Eq. (B.5), we arrive at

$$I_1' = \frac{z_1^{3-d}}{2}A'\zeta^{d-1}\left\{\Gamma\left(\frac{3}{2}\right)y^{\frac{3}{2}}\, {}_5\tilde{F}_4\left[\left\{\frac{3}{2}, \frac{5}{2}, \frac{d-1}{2}, \frac{d+3}{2}, \frac{d+7}{2}\right\}, \left\{\frac{5}{2}, 3, 5, \frac{d+4}{2}\right\}, y\right]\right\}_{\zeta^2}^1 \tag{B.11}$$

$$+ \frac{z_1^{3-d}}{2}B'\zeta^2\left\{\Gamma\left(\frac{d}{2}\right)y^{\frac{d}{2}}\, {}_5\tilde{F}_4\left[\left\{\frac{d}{2}, \frac{5}{2}, \frac{d-1}{2}, \frac{d+3}{2}, \frac{d+7}{2}\right\}, \left\{\frac{d+2}{2}, 3, 5, \frac{d+4}{2}\right\}, y\right]\right\}_{\zeta^2}^1$$

$$+ \frac{z_2^{3-d}}{2}A'\zeta^{1-d}\left\{\Gamma\left(\frac{d+4}{2}\right)y^{\frac{d+4}{2}}\, {}_5\tilde{F}_4\left[\left\{\frac{d+4}{2}, \frac{5}{2}, \frac{d-1}{2}, \frac{d+3}{2}, \frac{d+7}{2}\right\}, \left\{\frac{d+6}{2}, 3, 5, \frac{d+4}{2}\right\}, y\right]\right\}_0^{\zeta^2}$$

$$+ \frac{z_2^{3-d}}{2}B'\zeta^{-2}\left\{\Gamma\left(\frac{7}{2}\right)y^{\frac{7}{2}}\, {}_5\tilde{F}_4\left[\left\{\frac{7}{2}, \frac{5}{2}, \frac{d-1}{2}, \frac{d+3}{2}, \frac{d+7}{2}\right\}, \left\{\frac{9}{2}, 3, 5, \frac{d+4}{2}\right\}, y\right]\right\}_0^{\zeta^2}.$$

We skip the derivation for $I_2'$ for simplicity, as it is similar. The final result is given by

$$I_1' + I_2' = \frac{z_1^{3-d}}{2}A'\zeta^{d-1}\left\{\Gamma\left(\frac{3}{2}\right){}_5\tilde{F}_4\left[\left\{\frac{3}{2}, \frac{5}{2}, \frac{d-1}{2}, \frac{d+3}{2}, \frac{d+7}{2}\right\}, \left\{\frac{5}{2}, 3, 5, \frac{d+4}{2}\right\}, 1\right]\right. \tag{B.12}$$

$$\left. + \Gamma\left(\frac{d+4}{2}\right){}_5\tilde{F}_4\left[\left\{\frac{d+4}{2}, \frac{5}{2}, \frac{d-1}{2}, \frac{d+3}{2}, \frac{d+7}{2}\right\}, \left\{\frac{d+6}{2}, 3, 5, \frac{d+4}{2}\right\}, 1\right]\right\}$$

$$+ \frac{z_1^{3-d}}{2}B'\zeta^2\left\{\Gamma\left(\frac{d}{2}\right){}_5\tilde{F}_4\left[\left\{\frac{d}{2}, \frac{5}{2}, \frac{d-1}{2}, \frac{d+3}{2}, \frac{d+7}{2}\right\}, \left\{\frac{d+2}{2}, 3, 5, \frac{d+4}{2}\right\}, 1\right]\right.$$

$$\left. + \Gamma\left(\frac{7}{2}\right){}_5\tilde{F}_4\left[\left\{\frac{7}{2}, \frac{5}{2}, \frac{d-1}{2}, \frac{d+3}{2}, \frac{d+7}{2}\right\}, \left\{\frac{9}{2}, 3, 5, \frac{d+4}{2}\right\}, 1\right]\right\}$$

$$+ \frac{z_1^{3-d}}{2}A'\zeta^{d+2}\left\{\Gamma\left(\frac{d+4}{2}\right){}_5\tilde{F}_4\left[\left\{\frac{d+4}{2}, \frac{5}{2}, \frac{d-1}{2}, \frac{d+3}{2}, \frac{d+7}{2}\right\}, \left\{\frac{d+6}{2}, 3, 5, \frac{d+4}{2}\right\}, \zeta^2\right]\right.$$

$$\left. - \Gamma\left(\frac{3}{2}\right){}_5\tilde{F}_4\left[\left\{\frac{3}{2}, \frac{5}{2}, \frac{d-1}{2}, \frac{d+3}{2}, \frac{d+7}{2}\right\}, \left\{\frac{5}{2}, 3, 5, \frac{d+4}{2}\right\}, \zeta^2\right]\right\}$$

$$+ \frac{z_1^{3-d}}{2}B'\zeta^{d+2}\left\{\Gamma\left(\frac{7}{2}\right){}_5\tilde{F}_4\left[\left\{\frac{7}{2}, \frac{5}{2}, \frac{d-1}{2}, \frac{d+3}{2}, \frac{d+7}{2}\right\}, \left\{\frac{9}{2}, 3, 5, \frac{d+4}{2}\right\}, \zeta^2\right]\right.$$

$$\left. - \Gamma\left(\frac{d}{2}\right){}_5\tilde{F}_4\left[\left\{\frac{d}{2}, \frac{5}{2}, \frac{d-1}{2}, \frac{d+3}{2}, \frac{d+7}{2}\right\}, \left\{\frac{d+2}{2}, 3, 5, \frac{d+4}{2}\right\}, \zeta^2\right]\right\}.$$

Note that this is the result for $z_1 < z_2$. The result for $z_1 > z_2$ can be obtained by exchanging $z_1$ and $z_2$. With these integrals, the one-loop calculation for the layer susceptibility is

$$c_{23}' = \frac{90^{\frac{3}{2}}}{(6!)^2}\frac{S_{d-1}}{128(2\pi)^{d-1}}\sqrt{u_0}(z_1 z_2)^{\frac{4-d}{2}}\left(A'C_1'\zeta^{\frac{d+1}{2}} + B'C_2'\zeta^{\frac{7-d}{2}}\right) + (z_1 z_2)^{\frac{4-d}{2}}H'(\zeta), \tag{B.13}$$

where

$$C_1' = \Gamma\left(\frac{5}{2}\right)\Gamma\left(\frac{d-1}{2}\right)\Gamma\left(\frac{d+3}{2}\right)\Gamma\left(\frac{d+7}{2}\right) \tag{B.14}$$

$$\times \left\{\Gamma\left(\frac{3}{2}\right){}_5\tilde{F}_4\left[\left\{\frac{3}{2}, \frac{5}{2}, \frac{d-1}{2}, \frac{d+3}{2}, \frac{d+7}{2}\right\}, \left\{\frac{5}{2}, 3, 5, \frac{d+4}{2}\right\}, 1\right]\right.$$

$$\left. + \Gamma\left(\frac{d+4}{2}\right){}_5\tilde{F}_4\left[\left\{\frac{d+4}{2}, \frac{5}{2}, \frac{d-1}{2}, \frac{d+3}{2}, \frac{d+7}{2}\right\}, \left\{\frac{d+6}{2}, 3, 5, \frac{d+4}{2}\right\}, 1\right]\right\},$$

$$C_2' = \Gamma\left(\frac{5}{2}\right)\Gamma\left(\frac{d-1}{2}\right)\Gamma\left(\frac{d+3}{2}\right)\Gamma\left(\frac{d+7}{2}\right)$$
$$\times\left\{\Gamma\left(\frac{d}{2}\right){}_5\tilde{F}_4\left[\left\{\frac{d}{2},\frac{5}{2},\frac{d-1}{2},\frac{d+3}{2},\frac{d+7}{2}\right\},\left\{\frac{d+2}{2},3,5,\frac{d+4}{2}\right\},1\right]\right.$$
$$\left.+\Gamma\left(\frac{7}{2}\right){}_5\tilde{F}_4\left[\left\{\frac{7}{2},\frac{5}{2},\frac{d-1}{2},\frac{d+3}{2},\frac{d+7}{2}\right\},\left\{\frac{9}{2},3,5,\frac{d+4}{2}\right\},1\right]\right\},$$

and $H'(\zeta)$ is shown in Eq. (82) in the main text.

The evaluation for $I_{1,2}''$ is similar, so we skip the derivation but present the final results. The layer susceptibility $c_{24}'$ is

$$c_{24}' = \frac{90^{\frac{3}{2}}}{(6!)^2}\frac{S_{d-1}}{128(2\pi)^{d-1}}\sqrt{u_0}(z_1 z_2)^{\frac{4-d}{2}}\left(A'C_1''\zeta^{\frac{d+1}{2}}+B'C_2''\zeta^{\frac{7-d}{2}}\right)+(z_1 z_2)^{\frac{4-d}{2}}H''(\zeta), \tag{B.15}$$

where

$$C_1'' = \Gamma\left(\frac{3}{2}\right)\Gamma\left(\frac{d-1}{2}\right)\Gamma\left(\frac{d+1}{2}\right)\Gamma\left(\frac{d+3}{2}\right) \tag{B.16}$$
$$\times\left\{\Gamma\left(\frac{1}{2}\right){}_5\tilde{F}_4\left[\left\{\frac{1}{2},\frac{3}{2},\frac{d-1}{2},\frac{d+1}{2},\frac{d+3}{2}\right\},\left\{\frac{3}{2},2,3,\frac{d+2}{2}\right\},1\right]\right.$$
$$\left.+\Gamma\left(\frac{d+2}{2}\right){}_5\tilde{F}_4\left[\left\{\frac{d+2}{2},\frac{3}{2},\frac{d-1}{2},\frac{d+1}{2},\frac{d+3}{2}\right\},\left\{\frac{d+4}{2},2,3,\frac{d+2}{2}\right\},1\right]\right\},$$
$$C_2'' = \Gamma\left(\frac{3}{2}\right)\Gamma\left(\frac{d-1}{2}\right)\Gamma\left(\frac{d+1}{2}\right)\Gamma\left(\frac{d+3}{2}\right)$$
$$\times\left\{\Gamma\left(\frac{d-2}{2}\right){}_5\tilde{F}_4\left[\left\{\frac{d-2}{2},\frac{3}{2},\frac{d-1}{2},\frac{d+1}{2},\frac{d+3}{2}\right\},\left\{\frac{d}{2},2,3,\frac{d+2}{2}\right\},1\right]\right.$$
$$\left.+\Gamma\left(\frac{5}{2}\right){}_5\tilde{F}_4\left[\left\{\frac{5}{2},\frac{3}{2},\frac{d-1}{2},\frac{d+1}{2},\frac{d+3}{2}\right\},\left\{\frac{7}{2},2,3,\frac{d+2}{2}\right\},1\right]\right\},$$

and $H''(\zeta)$ is shown in Eq. (85) in the main text.

## C  Solution of the coupled RG equations Eq. (108)

In this section, we discuss the full solution of the RG equations, Eq. (108). Defining $\mu = e^{-t}$, we can solve the RG equations with the initial conditions $s(0) = s_0$ and $g(0) = g_0$:

$$s(t) = (4at + s_0^4)^{1/4}, \qquad g(t) = \frac{12ag_0}{\pi g_0(4at+s_0^4)^{3/2}-12abg_0 t+12a-\pi g_0 s_0^6}, \tag{C.1}$$

where $a = \frac{3(3N+22)}{128\pi^4}$ and $b = \frac{N-2}{2\pi}$ are two constants. It is then straightforward to classify different regions of the RG flow according to the exact solution. We introduce the following functions,

$$\tilde{g}_0 = \frac{12a\pi^2}{(\pi s_0^2-2b)^2(\pi s_0^2+b)}, \qquad \tilde{s}_0 = \sqrt{\frac{2b}{\pi}}. \tag{C.2}$$

We are ready to discuss the RG flow. Because the perturbative coupling is $s' = gs$, we plot the RG flow diagram of $g$-$s'$ in Fig. 4 (a) with $N = 6$. The flow diagram consists of three regions:

- $g_0 < \tilde{g}_0$, corresponds to the blue region, where the couplings flow to slightly larger values and back to $(0,0)$.

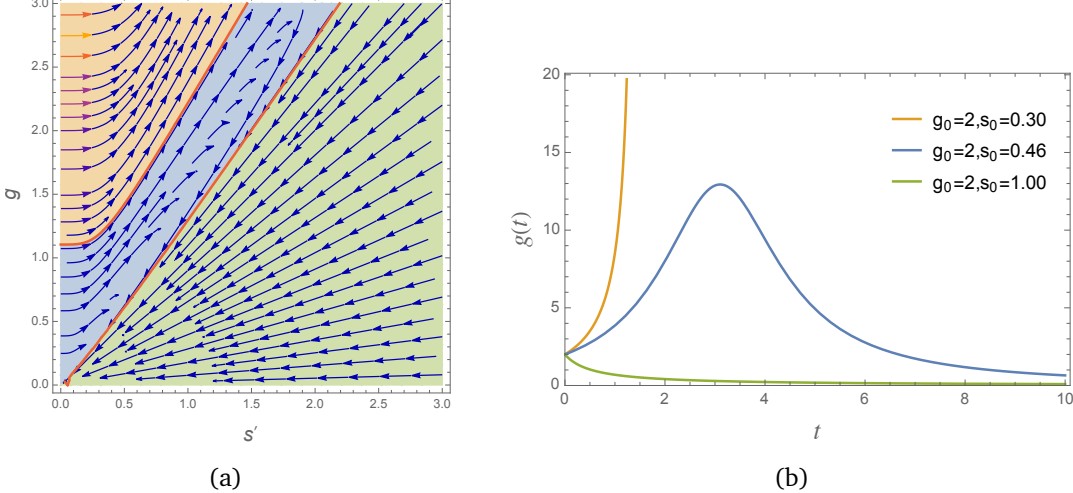

Figure 4: (a) The renormalization group (RG) flow diagram for the couplings $g$-$s'$ for $N = 6$. Three different regions are indicated by different colors. (b) The RG flow of the coupling $g$ in the three regions, with the curves plotted in corresponding colors.

- $g_0 > \tilde{g}_0$, $s_0 > \tilde{s}_0$, corresponds to the green region, where the couplings flow to $(0, 0)$.

- $g_0 > \tilde{g}_0$, $s_0 < \tilde{s}_0$, corresponds to the orange region, where the couplings flow to infinity. It is beyond the control of the perturbative method.

An example of the flow of the coupling $g$ is shown in Fig. 4 (b), illustrating the qualitative behavior in the three regions, with the curves plotted in corresponding colors. In the blue region, although the coupling $g$ initially increases, the flow ultimately approaches $(0, 0)$ in the deep infrared, indicating that the perturbative calculation remains self-consistent. This is further supported by the fact that as the initial value $g_0$ decreases, the maximum value reached by $g$ along the flow also decreases, keeping the entire trajectory within the perturbative regime. We therefore conclude that for sufficiently small $g_0$, the RG flow is controlled by the perturbative expansion and leads to a self-consistent fixed point.

# D  Correlation function of the critical $O(N)$ theory

In this section, we apply our general method to calculate the correlation function in the scalar $\phi^4$ theory. We focus on the most complicated Feynman diagram, which yields the same result as Eq. (3.15)-(3.16) in Ref. [26].

Beginning with Eq. (3.6) from Ref. [26], the layer susceptibility at the one-loop level reads

$$\chi(z, z') = \frac{u_0^2}{2} \int_0^\infty dy \int_0^\infty dy' \, G_0(p = 0; z, y) m_0(y) \left[ \int d^{d-1} r \, G_0^2(r, y, y') \right] m_0(y') G_0(p = 0; y', z'), \quad \text{(D.1)}$$

where $m_0(z) = \sqrt{\frac{12}{u_0}} \frac{1}{z}$. With $n = 2$ for the $\phi^4$ theory and $\alpha_L = \frac{3}{2} + 1 = \frac{5}{2}$ in Eq. (46), the mean-field propagator in the mixed representation becomes

$$G_0(p; z, z') = \sqrt{zz'} I_{5/2}(pz) K_{5/2}(pz'), \qquad z < z', \quad \text{(D.2)}$$

and $G_0(p = 0; z, z') = \frac{1}{5} \sqrt{zz'} \zeta^{\frac{5}{2}}$.

Therefore, using the same method as in the tricritical point, assuming $z < z'$ and $\zeta = \frac{z}{z'}$, we can evaluate the bubble,

$$
\begin{aligned}
b^L(z,z') &= \int \mathrm{d}^{d-1}r \; G_0^2(r,y,y') \\
&= \frac{S_{d-1}}{(2\pi)^{d-1}} z z' \frac{1}{4} z^5 z'^{-d-4} \Gamma\left(\frac{d-1}{2}\right) \Gamma(3) \Gamma\left(\frac{d+4}{2}\right) \Gamma\left(\frac{d+9}{2}\right) \\
&\quad \times {}_4\tilde{F}_3\left[\left\{\frac{d-1}{2}, 3, \frac{d+4}{2}, \frac{d+9}{2}\right\}, \left\{\frac{7}{2}, \frac{d+5}{2}, 6\right\}, \left(\frac{z}{z'}\right)^2\right].
\end{aligned}
\tag{D.3}
$$

Then plugging it into $\chi(z,z')$, we arrive at

$$
\chi(z,z') = \frac{6u_0}{25} \frac{1}{4} \frac{S_{d-1}}{(2\pi)^{d-1}} \Gamma\left(\frac{d-1}{2}\right) \Gamma(3) \Gamma\left(\frac{d+4}{2}\right) \Gamma\left(\frac{d+9}{2}\right) (I_1 + I_2),
\tag{D.4}
$$

where $I_{1,2}$, not shown explicitly, are integrals of regularized hypergeometric function, and can be done by using

$$
\int_0^\infty \mathrm{d}y \, y^{2-d} \frac{\min(z_1,y)^3}{\max(z_1,y)^2} \frac{\min(z_2,y)^3}{\max(z_2,y)^2} = z_2^{5-d}(A'' \zeta^{7-d} + B'' \zeta^3),
\tag{D.5}
$$

where $\zeta = \frac{z_1}{z_2}$ for $z_1 < z_2$, and $A'' = \frac{-5}{(4-d)(9-d)}$, $B'' = \frac{5}{(4-d)(1+d)}$.

Because $z_1$ and $z_2$ are symmetric, we can exchange them to get the result for $z_2 < z_1$. For brevity, we omit the tedious derivation. Finally, the layer susceptibility is

$$
\chi(z,z') = \frac{6u_0}{25} \frac{S_{d-1}}{(2\pi)^{d-1}} z'^{5-d} \zeta^3 \left(\frac{A''\tilde{C}_1}{8} \zeta^{4-d} + \frac{B''\tilde{C}_2}{8} + h(\zeta)\right),
\tag{D.6}
$$

where $\tilde{C}_1$ and $\tilde{C}_2$ are given, respectively, by

$$
\begin{aligned}
\tilde{C}_1 = {}&\Gamma\left(\frac{d-1}{2}\right) \Gamma(3) \Gamma\left(\frac{d+4}{2}\right) \Gamma\left(\frac{d+9}{2}\right) \\
&\times \left\{\Gamma\left(\frac{d}{2}\right) {}_5\tilde{F}_4\left[\left\{\frac{d}{2}, \frac{d-1}{2}, 3, \frac{d+4}{2}, \frac{d+9}{2}\right\}, \left\{\frac{d+2}{2}, \frac{7}{2}, \frac{d+5}{2}, 6\right\}, 1\right]\right. \\
&\left.+ \Gamma\left(\frac{9}{2}\right) {}_5\tilde{F}_4\left[\left\{\frac{9}{2}, \frac{d-1}{2}, 3, \frac{d+4}{2}, \frac{d+9}{2}\right\}, \left\{\frac{11}{2}, \frac{7}{2}, \frac{d+5}{2}, 6\right\}, 1\right]\right\},
\end{aligned}
\tag{D.7}
$$

$$
\begin{aligned}
\tilde{C}_2 = {}&\Gamma\left(\frac{d-1}{2}\right) \Gamma(3) \Gamma\left(\frac{d+4}{2}\right) \Gamma\left(\frac{d+9}{2}\right) \\
&\times \left\{\Gamma(2) {}_5\tilde{F}_4\left[\left\{2, \frac{d-1}{2}, 3, \frac{d+4}{2}, \frac{d+9}{2}\right\}, \left\{3, \frac{7}{2}, \frac{d+5}{2}, 6\right\}, 1\right]\right. \\
&\left.+ \Gamma\left(\frac{d+5}{2}\right) {}_5\tilde{F}_4\left[\left\{\frac{d+5}{2}, \frac{d-1}{2}, 3, \frac{d+4}{2}, \frac{d+9}{2}\right\}, \left\{\frac{d+7}{2}, \frac{7}{2}, \frac{d+5}{2}, 6\right\}, 1\right]\right\},
\end{aligned}
\tag{D.8}
$$

and the function $h(\zeta)$ is given explicitly by

$$
\begin{aligned}
h(\zeta) = {}&\frac{1}{4}\Gamma\left(\frac{d-1}{2}\right) \Gamma(3) \Gamma\left(\frac{d+4}{2}\right) \Gamma\left(\frac{d+9}{2}\right) \\
&\times \left(\frac{A''}{2} \zeta^4 \Gamma\left(\frac{9}{2}\right) {}_5\tilde{F}_4\left[\left\{\frac{9}{2}, \frac{d-1}{2}, 3, \frac{d+4}{2}, \frac{d+9}{2}\right\}, \left\{\frac{11}{2}, \frac{7}{2}, \frac{d+5}{2}, 6\right\}, \zeta^2\right]\right. \\
&+ \frac{B''}{2} \zeta^4 \Gamma\left(\frac{d+5}{2}\right) {}_5\tilde{F}_4\left[\left\{\frac{d+5}{2}, \frac{d-1}{2}, 3, \frac{d+4}{2}, \frac{d+9}{2}\right\}, \left\{\frac{d+7}{2}, \frac{7}{2}, \frac{d+5}{2}, 6\right\}, \zeta^2\right] \\
&- \frac{A''}{2} \zeta^4 \Gamma\left(\frac{d}{2}\right) y^{\frac{d}{2}} {}_5\tilde{F}_4\left[\left\{\frac{d}{2}, \frac{d-1}{2}, 3, \frac{d+4}{2}, \frac{d+9}{2}\right\}, \left\{\frac{d+2}{2}, \frac{7}{2}, \frac{d+5}{2}, 6\right\}, \zeta^2\right] \\
&\left.- \frac{B''}{2} \zeta^4 \Gamma(2) {}_5\tilde{F}_4\left[\left\{2, \frac{d-1}{2}, 3, \frac{d+4}{2}, \frac{d+9}{2}\right\}, \left\{3, \frac{7}{2}, \frac{d+5}{2}, 6\right\}, \zeta^2\right]\right).
\end{aligned}
\tag{D.9}
$$

To compare the results with Eqs. (3.15) and (3.16) in Ref. [26], we need to simplify $\tilde{C}_1$ and $\tilde{C}_2$ in Eq. (D.7) and Eq. (D.8). To expand them around $d = 4 - \epsilon$, we rewrite them in terms of a summation,

$$
\begin{aligned}
\tilde{C}_1 = {}& 2^{6-d} \sum_{k=0}^{\infty} \left[ \frac{(k+1)(k+\frac{d+7}{2})(k+\frac{d+5}{2})(k+\frac{d+9}{2})(k+\frac{d+2}{2})}{(k+5)(k+4)(k+3)(k+\frac{5}{2})(k+\frac{9}{2})} - 1 \right] \frac{\Gamma(2k+d-1)}{\Gamma(2k+4)} \\
& + 2^{6-d} \sum_{k=0}^{\infty} \frac{\Gamma(2k+d-1)}{\Gamma(2k+4)} \,.
\end{aligned}
\tag{D.10}
$$

We separate the expression into two parts: the first part remains finite for $\epsilon \to 0$, while the second part diverges as $2^{6-d} \sum_{k=0}^{\infty} \frac{\Gamma(2k+d-1)}{\Gamma(2k+4)} \approx \frac{2}{\epsilon}$. While not shown here, the leading term of $\tilde{C}_2$ can be obtained similarly.

In Ref. [26], for $z < z'$, they gave the layer susceptibility as follows,

$$
\begin{aligned}
\chi(z,z') = {}& \frac{1}{5} \frac{z^3}{z'^2} \frac{24}{5} u_0 \frac{S_d^{-1}}{d-2} 2^{2-d} z'^{4-d} \left[ \frac{\zeta^{4-d}K}{9-d} + \frac{L - \zeta^{4-d}K}{4-d} + \frac{L}{d+1} + H_d(\zeta) \right] \\
= {}& \frac{6}{25} u_0 z'^{5-d} \zeta^3 \frac{S_{d-1}}{(2\pi)^{d-1}} 2^{3-d} \Gamma(d-2) \left[ A'' K \zeta^{4-d} + B'' L + H_d(\zeta) \right],
\end{aligned}
\tag{D.11}
$$

where we used $\frac{S_d^{-1}}{d-2} 2^{4-d} \cdot \frac{(2\pi)^{d-1}}{S_{d-1}} = \frac{2^{3-d}}{d-2} \Gamma(d-1)$. $A''$ and $B''$ are defined in Eq. (D.5), and $K$ and $L$ are

$$
K = \frac{(2-\epsilon)(72-\epsilon^2)}{6\epsilon(2+\epsilon)(4+\epsilon)(6+\epsilon)}, \qquad L = \frac{(2-\epsilon)(4-\epsilon)(6-\epsilon)}{48\epsilon(2+\epsilon)},
\tag{D.12}
$$

with $\epsilon = 4 - d$. The function $H_d(\zeta)$ is a complicated expression, and is not shown here.

To prove the equivalence of two results, we need to check that $2^{3-d}\Gamma(d-2)K = \frac{\tilde{C}_1}{8}$, $2^{3-d}\Gamma(d-2)L = \frac{\tilde{C}_2}{8}$ and $2^{3-d}\Gamma(d-2)H_d(\zeta) = h(\zeta)$. Although it is challenging to prove them for general $d$, we can show that $2^{3-d}\Gamma(d-2)K = \frac{\tilde{C}_1}{8}$ and $2^{3-d}\Gamma(d-2)L = \frac{\tilde{C}_2}{8}$ are valid up to $\mathcal{O}(\epsilon^2)$. Also, $2^{3-d}\Gamma(d-2)H_d(\zeta) = h(\zeta)$ is valid up to $\mathcal{O}(\zeta^{22})$. It would be interesting to give a rigorous proof of these relations in the future.

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
