# Peer review of "Boundary operator expansion and extraordinary phase transition in the tricritical O(N) model"

_SciPost Physics, doi:SciPost Phys. 18, 210 (2025)_

## Round 1 · Referee Report · Mykola A. Shpot (Referee 1) · 2025-3-21

Report

The preprint by Xinyu Sun and Shao-Kai Jian belongs to the series of contemporary papers written by young authors who try to discover and theoretically describe certain unusual phenomena that can happen in semi-infinite systems at bulk criticality, especially within their surface layer.
The authors aim to find a possibly new boundary critical behavior at the bulk tricritical point. The significance of the manuscript is in that they seem to have succeeded. The idea of studying the semi-infinite tricritical systems is not new, but the originality of the manuscript is provided by applying to the problem the methods developed quite recently by Dey, Hansen and Shpot, and Metlitski.
The manuscript is original, up-to-date, scientifically sound and interesting, and thus deserves publication in SciPost Physics. It can be accepted with modifications implied by the comments given in the attached report.

Attachment

Recommendation

Ask for minor revision

  • validity: -
  • significance: -
  • originality: -
  • clarity: -
  • formatting: -
  • grammar: -

Author:  Xinyu Sun  on 2025-05-23  [id 5511]

(in reply to Report 1 by Mykola A. Shpot on 2025-03-21)

Thank you for the thoughtful and constructive comments on my manuscript. I have carefully considered the points raised in the Report, and I outline my detailed responses in the attached document.

Attachment:

reply_to_Report_1.pdf

---

## Round 1 · Referee Report · Anonymous (Referee 2) · 2025-3-25

Strengths

  1. Computes previously unreported properties of the tricritical O(N) model.

  2. Discovers nontrivial surface ordering in d=3 in a less contrived scenario than those in Ref. 22

  3. Gives convincing argument why this ordering occurs.

Weaknesses

  1. Level of grammar and precision in language is somewhat lacking.

  2. Introduction does not sufficiently motivate the paper.

Report

This paper's content is well suited for the journal. However, the paper could use improvements in precision of language, uniformity of language, grammatical content, and motivation in the introduction. I have provided some unclear areas that the other referee did not mention (but both set of areas should be addressed). After making these changes, I would recommend the publication of this article.

Requested changes

  1. The sentence in the abstract "Then, by employing the technique of layer susceptibility, we solve the boundary operator expansion using the $\epsilon = 3-d$ expansion" is unclear. Layer susceptibility is not a technique. Moreover, the sentence does not specify which BOE is computed.

  2. The introduction spends an entire paragraph on using layer susceptibility to compute BOE. I would not state that this is one of the main innovations of the paper, so perhaps this calculation technique should not be presented as such.

  3. On the other hand, the introduction should provide physical motivation for studying the tricritical O(N) model (e.g., the \phi^6 term being marginal at d=3 -- this could affect the RG flow and boundary behavior substantially). The paper should also emphasize Ref. 22 more in the introduction rather than discussing their result in the context of Ref. 22 as an aside before the conclusion.

  4. The sentence "Broadly, we believe... systems" likely belongs in the conclusion rather than the introduction. Moreover, the authors do not justify why this is true.

  5. The allowed values of $\Delta$ should be specified in equations 1 and 2 or in future mentions of the BOE.

  6. Better notation for $G^T$ on page 4 is $<\phi^i \phi^j> = \delta_{ij} G^T$.

  7. The discussion of $c_0$ after equation 8 should be unified with the discussion of $c_0$ after equation 6 -- we already take $c_0 \to- \infty$ after equation 6.

  8. What is $k$ in Equation 18? Errors like these (that the other referee also brought up) should be proofed.

  9. Minor formatting point, the parentheses in Equation 28 and other places should fit the expressions.

  10. The authors are not consistent with their use of tense. E.g., on page 7, the authors use future tense and present tense to describe results in the paper. Elsewhere, the authors use past tense to also describe results in the paper. I recommend using past tense for prior work and present tense for results in the paper.

  11. The Green's functions in equation 31 are not labeled by longitudinal or transverse. The word "respectively" should be added after "longitudinal and transverse fields" in the sentence after this equation.

  12. As an example of grammar that needs to be fixed, on page 14, the penultimate sentence in the paragraph after Equation 65 should read "can be found in Appendix B.1."

  13. Likewise, the sentence after Equation 87 should be proofed, and later on that page, the authors misspell "resumming."

  14. I agree with the other referee's comments on the clarity of the derivation of the extraordinary-log transition. Ref 19's argument might be easier to use for this derivation.

  15. I am quite confused by the assignment of a critical $N_c$ in equation 106. It seems as if the authors are treating the boundary as still being a 2d plane for $d = 3-\epsilon$. If the authors treated the boundary as a $2-\epsilon$d plane, a linear term would appear in the RG equation for $g$, and so the extraordinary-log universality class would not exist. Is treating the boundary as a 2d plane physically meaningful here? A codimension $1-\epsilon$ defect seems odd to treat.

  16. I agree with the other referee on the spelling of "Callan-Symanzik". However, I think using "Callan-Symanzik" in this context is as appropriate as using "renormalization group equation" -- I have seen both conventions used in the literature.

  17. As I mentioned earlier, the last paragraph on page 24 should be emphasized a lot more.

Recommendation

Ask for minor revision

  • validity: top
  • significance: high
  • originality: good
  • clarity: ok
  • formatting: reasonable
  • grammar: below threshold

Author:  Xinyu Sun  on 2025-05-23  [id 5512]

(in reply to Report 2 on 2025-03-25)

I appreciate the valuable feedback provided in the Report. Please find my detailed responses to each of the comments in the attached file.

Attachment:

reply_to_report_2.pdf

---

## Round 1 · Referee Report · Anonymous (Referee 3) · 2025-3-26

Strengths

  1. The manuscript gives an important physical result and uses appropriate evidence to justify the claim.
  2. The calculations are thorough and the results are convincing.
  3. This paper proves that there is an extraordinary transition in the tricritical O(N) models in 3d, which is a significant result that can potentially be observed in experiments and/or numerical simulations

Weaknesses

  1. The presentation of the paper could be improved, and the main result can be emphasized more (see report).
  2. There are many grammatical errors and typos in the manuscript as it stands, which need to be corrected.

Report

The presented manuscript studies the extraordinary transition for the tricritical O(N) model using RG. The authors calculate the correlators (and layer susceptibilities) upto one loop order in the d = 3 - \epsilon expansion (the upper critical dimension is 3). Using the correlators, the authors are able to extract various CFT data including the boundary OPE coefficients for the one-point functions of displacement and tilt. The authors then go on to show using an RG argument that the epsilon expansion results imply that the tricritical O(N) model has an extraordinary transition where the bulk undergoes a phase transition in the presence of an ordered surface. This is interesting and relevant because the Mermin-Wagner theorem prevents the spontaneous symmetry breaking of continuous symmetries in 2d, but we can see that the O(N) symmetry can be spontaneously broken on the 2d boundary of a 3d system. The methods used in the work are traditional, and well established. It is justified to use the epsilon expansion here because unlike the regular O(N) models the upper critical dimension here is 3, not 4.

My only minor qualm is with the structure of the paper, which I leave for the authors to ponder. I think the reading experience and flow will be enhanced if section 3 is relegated to an appendix. The calculations are essential for what comes after, yes, but it also makes for long equations spanning half a page. The physical results are already emphasized in section 1.1. The technical section 3 might deter readers from appreciating the important discussions in sections 4 & 5.

Finally, the grammar of the paper needs improving. I found many errors and typos while reading, some of which I point out in the list of changes. Please attend to them and proofread the manuscript once again. After the grammatical errors are ironed out, I think this paper would be very suitable for publishing in SciPost Phys. The content is original and interesting, and the derivations support the claim.

Requested changes

  1. Abstract and elsewhere - "2D dimensions" should be replaced by either "2d" or "two-dimensions"
  2. In the first paragraph of introduction, the word "enhance" reads weird, as in "the presence of a boundary enhances the bulk CFT". Perhaps it should say imbibes additional structure.
  3. Eq. (1), c_\delta not defined. I suggest moving the line below Eq. (3) higher, below Eq. (2).
  4. Eq. (4), p is not defined around this equation, although clear from context that it is the momentum in directions parallel to the boundary.
  5. In Section 5, the line above Eq. (93), "couple the ordinary transition action to the nonlinear sigma model according to O(N) symmetry" should be "restoring/respecting O(N) symmetry"
  6. In the paragraph above Eq. (95), the authors state "The normal boundary condition is actually equivalent to the ordinary transition". This is not correct, atleast for the O(N) models. In this case, after doing the entire RG analysis, we realize that the boundary is ordered. But a priori these are different boundary conditions and should be treated as such. Extraordinary transition is the spontaneous symmetry breaking of O(N) symmetry whereas normal boundary condition explicitly breaks O(N) symmetry by an external field.

Recommendation

Ask for minor revision

  • validity: top
  • significance: good
  • originality: high
  • clarity: good
  • formatting: good
  • grammar: below threshold

Author:  Xinyu Sun  on 2025-05-23  [id 5513]

(in reply to Report 3 on 2025-03-26)

Thank you for your careful review and insightful comments. I have addressed each point raised in the Report in detail. Please refer to the attached document for my full responses.

Attachment:

reply_to_report_3.pdf

---

## Round 2 · Author Response

List of changes

---

## Round 2 · List of Changes



---

## Editorial Decision

published